# Using whole-genome sequence data to examine the epidemiology of Salmonella, Escherichia coli and associated antimicrobial resistance in raccoons (Procyon lotor), swine manure pits, and soil samples on swine farms in southern Ontario, Canada

Nadine A. Vogt[1]*, Benjamin M. Hetman[1], David L. Pearl[1], Adam A. Vogt[2], Richard J. Reid-Smith[1,3], E. Jane Parmley[1,3], Nicol Janecko[4], Amrita Bharat[5,6], Michael R. Mulvey[5,6], Nicole Ricker[7], Kristin J. Bondo[7¤], Samantha E. Allen[7,8,9], Claire M. Jardine[7,10]

**1** Department of Population Medicine, Ontario Veterinary College, Guelph, Ontario, Canada, **2** Independent Researcher, Mississauga, Ontario, Canada, **3** Centre for Foodborne, Environmental and Zoonotic Infectious Diseases, Public Health Agency of Canada, Guelph, Ontario, Canada, **4** Quadram Institute Bioscience, Norwich, United Kingdom, **5** National Microbiology Laboratory, Public Health Agency of Canada, Winnipeg, Manitoba, Canada, **6** Department of Medical Microbiology and Infectious Diseases, University of Manitoba, Winnipeg, Manitoba, Canada, **7** Department of Pathobiology, Ontario Veterinary College, Guelph, Ontario, Canada, **8** Wyoming Game and Fish Department, Laramie, Wyoming, United States of America, **9** Department of Veterinary Sciences, University of Wyoming, Laramie, Wyoming, United States of America, **10** Canadian Wildlife Health Cooperative, Ontario Veterinary College, Guelph, Ontario, Canada

¤ Current address: Pennsylvania Cooperative Fish and Wildlife Unit, The Pennsylvania State University, University Park, Pennsylvania, United States of America

* nvogt@uoguelph.ca

## Abstract

To better understand the contribution of wildlife to the dissemination of Salmonella and antimicrobial resistance in Salmonella and Escherichia coli, we examined whole-genome sequence data from Salmonella and E. coli isolates collected from raccoons (Procyon lotor) and environmental sources on farms in southern Ontario. All Salmonella and phenotypically resistant E. coli collected from raccoons, soil, and manure pits on five swine farms as part of a previous study were included. We assessed for evidence of potential transmission of these organisms between different sources and farms utilizing a combination of population structure assessments (using core-genome multi-locus sequence typing), direct comparisons of multi-drug resistant isolates, and epidemiological modeling of antimicrobial resistance (AMR) genes and plasmid incompatibility (Inc) types. Univariable logistic regression models were fit to assess the impact of source type, farm location, and sampling year on the occurrence of select resistance genes and Inc types. A total of 159 Salmonella and 96 resistant E. coli isolates were included. A diversity of Salmonella serovars and sequence types were identified, and, in some cases, we found similar or identical Salmonella isolates and resistance genes between raccoons, soil, and swine manure pits. Certain Inc types and resistance genes associated with source type were consistently more likely to be identified

**Data Availability Statement:** All data files are available from the Scholars Portal Dataverse, VI: http://dx.doi.org/10.5683/SP/BLFRK5, and from the Agri-environmental Research Data Repository: http://dx.doi.org/10.5887/AERDR/10864/12074. All sequence data has been deposited to Genbank, and is available under BioProject PRJNA745182.

**Funding:** Funding was provided by the Ontario Ministry of Agriculture, Food, and Rural Affairs (OMAFRA), through the Ontario Agri-Food Innovation Alliance (UofG2016-2642). NAV received stipend funding through the Ontario Veterinary College, the University of Guelph, and a National Sciences and Engineering Research Council Postgraduate Scholarship-Doctoral. The funders had no role in study design, data collection and analysis, decision to publish, or preparation of the manuscript.

**Competing interests:** The authors have declared that no competing interests exist.

in isolates from raccoons than swine manure pits, suggesting that manure pits are not likely a primary source of those particular resistance determinants for raccoons. Overall, our data suggest that transmission of *Salmonella* and AMR determinants between raccoons and swine manure pits is uncommon, but soil-raccoon transmission appears to be occurring frequently. More comprehensive sampling of farms, and assessment of farms with other livestock species, as well as additional environmental sources (e.g., rivers) may help to further elucidate the movement of resistance genes between these various sources.

## Introduction

The rise of antimicrobial resistance (AMR) is a major global threat to the health of humans and animals alike [1, 2]. There is mounting evidence of widespread movement of AMR determinants (e.g., genes and the plasmids associated with their movement) within natural environments [3–5], and genes conferring resistance to high-priority antimicrobials (e.g., *mcr-1*) have been identified in avian and mammalian wildlife across the world [6–8]. It is generally recognized that wild animals may act as sentinels of environmental AMR pollution, but recent work suggests that wildlife may also physically disseminate AMR determinants from one location to another through their feces [9, 10]. With recognition of the importance of One Health approaches that consider different sampling sources, there is a need to integrate epidemiological investigations with new technologies such as whole-genome sequencing (WGS), which permit the assessment of the genetic basis of AMR at a higher resolution [11, 12].

A number of investigations combining genomics with epidemiology to examine foodborne pathogens and/or AMR in wildlife have recently been performed [9, 10, 13–15]. With much of the literature on AMR and wildlife focused on wild birds [16], mammalian wildlife such as raccoons (*Procyon lotor*) arguably merit further examination in this context due to their prevalent populations, tendency to forage in anthropogenic environments, and general proximity to human and domestic animal settings [17]. This work is part of a larger repeated cross-sectional study of wild meso-mammals (raccoons, primarily) on swine farms and conservation areas in southern Ontario, Canada between 2011 and 2013 [18–20]. The study region, within the Grand River Watershed (6800km$^2$), includes intensive agricultural activities, and a population of ~1 million people, providing us with an opportunity to examine the intersection between wildlife, livestock, and environmental sources in a heavily populated region of southern Ontario, Canada. Although the overall prevalence of *Salmonella* and phenotypic resistance among *E. coli* isolated from wildlife and soil samples in this previous study did not differ significantly between swine farms and conservation areas, certain strains (e.g., *Salmonella* Typhimurium var. Copenhagen DT104), serovars (e.g., *Salmonella* Agona) and resistance patterns appeared only in samples obtained on swine farms [18, 19]. In certain cases, it was unclear if the molecular determinants of resistance were shared or were distinct between different compartments of this environment, even when phenotypic resistance was the same.

Consequently, the primary objective of this study was to examine in detail the subset of samples obtained from swine farms in this previous study, applying WGS data from *Salmonella* and phenotypically resistant *E. coli* isolates to assess for evidence of potential transmission of these organisms and associated AMR determinants among raccoons, swine manure pits, and soil, on and between farms. To address this objective, a combination of population structure assessments, epidemiological modeling of select AMR determinants (i.e., genes, predicted plasmids), and direct comparisons of multidrug resistant isolates was performed. For

assessments of potential transmission utilizing statistical modeling, our aim was to determine the impact of source type, farm location, and sampling year on the occurrence of AMR determinants (identified *in silico*). A secondary objective of this work was to examine the validity of genotypic AMR identification using WGS data by calculation of test sensitivity and specificity, using prior phenotypic AMR data as the gold standard.

## Methods

### Sample collection

Samples for this study were previously collected as part of a repeated cross-sectional study of raccoons on swine farms and conservation areas in southern Ontario between 2011 and 2013 [18, 19]. For the present study, we included only isolates originating from swine farm environments. Sampling sources included raccoons, swine manure pits, and soil. The study region and sampling methods have been previously described and are available in Bondo et al. [18, 19]. Briefly, samples were obtained from monthly sampling of five swine farms in the Grand River watershed, near the cities of Guelph and Cambridge in Ontario, Canada. Raccoons were live-trapped, and animals were chemically immobilised to obtain a rectal fecal swab using a Cary-Blair applicator (BBL CultureSwab, Bd; Becton, Dickinson and Company, Maryland, USA). Individual animals were ear-tagged and microchipped for subsequent identification, and animals were sampled up to once monthly, but animals recaptured within the same trapping month were released immediately. Swine manure pit and soil samples were obtained from each site at the beginning of each trapping week. For soil samples, 10 g of soil was collected from within a 2-m radius of each animal trap and stored in a sterile container. Swine manure pits were sampled by pooling samples from two different depths (i.e., top 1/3, and mid-depth) at three different locations around the pit. All samples were kept on ice in the field until further processing.

### Previous culture and susceptibility testing

Samples were previously cultured for *Salmonella* and *E. coli* within three days of collection at the McEwen Group Research Group Lab at the Canadian Research Institute for Food Safety, University of Guelph (Guelph, Ontario, Canada) using standard culture-based methodology as previously described [18, 19]. One isolate of *Salmonella* and one isolate of *E. coli* from each sample was sub-cultured and tested further. Isolates were confirmed by biochemical testing and submitted for phenotypic susceptibility testing to the Antimicrobial Resistance Reference Laboratory (National Microbiology Laboratory (NML) at Guelph, Public Health Agency of Canada, Guelph, Ontario, Canada). Testing was completed in accordance with methods outlined by the Canadian Integrated Program for Antimicrobial Resistance Surveillance (CIPARS) [21]. Isolates were previously tested using the National Antimicrobial Resistance Monitoring System (NARMS; Sensititre, Thermo Scientific), and antimicrobial panel CMV3AGNF, which included the following 15 antimicrobials: gentamicin (GEN), kanamycin (KAN), streptomycin (STR), amoxicillin-clavulanic acid (AMC), cefoxitin (FOX), ceftiofur (TIO), ceftriaxone (CRO), ampicillin (AMP), chloramphenicol (CHL), sulfisoxazole (SOX), trimethoprim-sulfamethoxazole (SXT), tetracycline (TCY), nalidixic acid (NAL), ciprofloxacin (CIP), and azithromycin (AZM).

### Selection of isolates for whole-genome sequencing

All *Salmonella* isolates originating from swine farms were selected for sequencing and inclusion in this study. Due to resource constraints, only *E. coli* isolates demonstrating phenotypic

resistance to at least one of 15 antimicrobials examined were selected for sequencing and included in the present study. During 2011, three different *E. coli* isolates were cultured from each sample; for samples with more than one isolate demonstrating phenotypic resistance, a random number generator was used to select one resistant isolate for sequencing.

## DNA extraction, whole-genome sequencing and genome assembly

Cultures of *Salmonella* and *E. coli* were grown on Mueller Hinton Agar and incubated overnight at 35˚C. Cultures were then distributed to the NML (Public Health Agency of Canada) in Winnipeg for DNA extraction and short-read sequencing, or these steps were performed on site, at the University of Guelph and the NML in Guelph, Ontario, respectively. Genomic DNA extraction was performed using 1 ml of culture as input to the Qiagen DNEasy plant and tissue 96 extraction kit, according to manufacturer protocols (Qiagen, Hilden, Germany). Sequencing was then performed at the NML in Guelph or in Winnipeg, using Nextera XT library preparation and Illumina MiSeq version 3 (600-cycle kit) or NextSeq550 platforms, according to manufacturer protocols. Assembly of raw reads was performed using SPAdes [22], as part of the Shovill pipeline (version 1.0.1; https://github.com/tseemann/shovill) using the following settings: "—minlen 200—mincov 2;—assembler spades;—trim".

## Analysis of whole-genome assemblies

Prediction of legacy multi-locus sequence types was performed using MLST (version 2.19.0; https://github.com/tseemann/mlst) according to the Achtman 7-loci scheme for *Salmonella enterica* and *E. coli* (https://pubmlst.org/mlst/). Isolates were also typed using *fsac* (version 1.2.0; https://github.com/dorbarker/fsac) according to the core-genome multi-locus sequence typing (cgMLST) schemes available from Enterobase (https://enterobase.warwick.ac.uk/) with 3002- and 2513-loci schemes for *Salmonella* and *E. coli*, respectively. Isolates with 25 or more missing cgMLST loci were considered poor quality and excluded from any further analyses. Allelic differences between cgMLST profiles from different sources were calculated using R (version 3.6.3), and minimum spanning trees were created to provide visual representations of population structure based on cgMLST data [23]. Minimum spanning trees were created using the standalone GrapeTree software package (version 1.5) [23] and the "MSTreeV2" algorithm, which accounts for missing data. For minimum spanning trees visualizing overall populations of *Salmonella* and *E. coli*, lenient clustering thresholds (k>20 allelic differences) were used to provide a qualitative assessment of overlap between isolates from different sources, while minimizing unnecessary noise. A similar approach was used to construct a minimum spanning tree for isolates of *Salmonella* serovars common to both raccoons and swine manure pits; however, a more stringent clustering threshold was applied (k<10 allelic differences) to reflect a higher degree of similarity between those isolates. This threshold of 10 allelic differences is also consistent with the strain-level threshold used by PulseNet for *Salmonella* [24], thereby ensuring this latter minimum spanning tree was less prone to clustering of isolates from potentially different strains. Serotyping of *E. coli* isolates was performed using ECTyper (version 1.0.0, database version 1.0; https//github.com/phac-nml/ecoli_serotyping) and default settings. Serotyping of *Salmonella* isolates was performed using SISTR (version 1.1.1; https://github.com/phac-nml/sistr_cmd), and default settings with the "centroid" allele database.

Acquired resistance genes were identified using Abricate (version 0.8.13; https://github.com/tseemann/abricate) and the Resfinder database (current as of May-17-2020), with settings of 90% identity and 60% coverage. Identity and coverage settings were increased to 100% and 90%, respectively, to identify acquired beta-lactamases. Identification of plasmid incompatibility (Inc) types was performed using Abricate (version 0.8.13; https://github.com/tseemann/

abricate) and the Plasmidfinder database (current as of May-17-2020). Settings of 98% identity and 70% coverage were used.

For multidrug resistant *Salmonella* isolates containing the same, or similar, phenotypic resistance patterns and representing identical sequence types, Snippy (version 4.4.0; https://github.com/tseemann/snippy) was used to further distinguish genetic differences based on single-nucleotide polymorphisms (SNPs) using the entire genome. Isolate raw reads were used, along with default settings. Reference genomes used were: *Salmonella enterica* subsp. *enterica* serovar Typhimurium str. LT2 (Accession No.: NC_003197.2); *Salmonella enterica* subsp. *enterica* serovar Hadar str. RI_05P066 (Accession No.: ABFG00000000.1).

## Sensitivity and specificity of in silico AMR prediction

The sensitivity and specificity of *in silico* AMR prediction were calculated by antimicrobial class and overall (i.e., pooling all individual test results); phenotypic AMR results were considered the gold standard, and the presence of genotypic resistance was considered a positive test result. Isolates with intermediate susceptibility were categorized as susceptible. We elected not to assess test sensitivity or specificity of drug classes for which chromosomal mutations are known to confer a considerable proportion of expressed resistance (i.e., quinolones) [25]. As a quality control measure, isolates with missing genotypes for resistant phenotypic test results for three or more of the seven antimicrobial classes were examined and excluded from further analyses if they were also missing greater than 20 loci based on cgMLST.

## Statistical analyses

Univariable multi-level logistic regression was used to model the odds of identifying select Inc types and AMR genes found in *E. coli* and *Salmonella* from different sources. All statistical analyses were performed using STATA (STATA Intercooled 14.2; StataCorp, College Station, Texas, USA). Only Inc types and resistance genes with a prevalence greater than 10% and less than 90% were modeled. The following independent variables were examined: year of sampling, farm location (farm sites 6–10, as in Bondo et al. [18]), and source type (i.e., raccoon, swine manure pit, soil). Due to low effective samples sizes, univariable logistic regression was performed, with a random intercept to account for clustering of isolates obtained from the same raccoon or swine manure pit. For models that did not converge using the 'melogit' command, the model was subsequently fit using the 'meqrlogit' command, which uses QR decomposition of the variance-components matrix. Variance components were used to calculate intraclass correlation coefficients (ICCs) using the latent variable technique [26]. The fit of multi-level models was assessed by examining the best linear unbiased predictions (BLUPS) for normality and homoscedasticity, and Pearson's residuals were examined for outliers. If variance components were very small ($<1x10^{-3}$), the Bayesian information criterion (BIC) was used to compare the fit of the multi-level logistic regression model with an ordinary logistic regression; the better fitting model was reported [26]. If low effective sample sizes posed estimation issues for univariable models, exact logistic regression was used, and the score method was used to calculate *p*-values for these models. A significance level of $\alpha = 0.05$ was used, and all tests were two-tailed.

# Results

## Dataset

**Salmonella.**   Based on our study criteria, a total of 159 *Salmonella* isolates from the following sources were included: raccoon (n = 92), soil (n = 46), and swine manure pit (n = 21).

**Table 1. Resistance genes and plasmid incompatibility (Inc) groups identified *in silico* among phenotypically resistant *Salmonella enterica* from raccoons, swine manure pits, and soil samples on swine farms in southern Ontario, Canada 2011–2013 (n = 6/159).**

| NCBI accession number | Isolate id | Sequence type | Serovar | Source | Phenotypic resistance pattern[a] | Resistance genes | Inc group | Year | Farm location |
|---|---|---|---|---|---|---|---|---|---|
| JAIHBY000000000 | N18-00467 | **ST309** | Kiambu | Soil | AMP-TCY | $bla_{TEM-1}$, tet(A) | IncX1 | 2013 | 6 |
| JAIHBV000000000 | N18-00464 | **ST19** | Typhimurium | Raccoon | STR-SOX-TCY | aadA2, sul1, tet(A) | ColRNAI, IncFIIS, IncFIBS, Col156, Col440I | 2013 | 7 |
| JAIHBU000000000 | N18-00463 | **ST19** | Typhimurium | Manure | STR-SOX-TCY | aadA2, sul1, tet(A) | ColRNAI, IncFIIS, IncFIBS, Col156 | 2013 | 9 |
| JAIHBW000000000 | N18-00465 | **ST96** | Schwarzen-grund | Raccoon | STR-SOX-TCY | aadA4, sul1, tet(B) | IncHI2, pkpccav1321, IncHI2A | 2013 | 6 |
| JAIHBX000000000 | N18-00466 | **ST33** | Hadar | Soil | STR-TCY | aph(3')-Ib, aph(6)-Id, tet(A) | ColRNAI, ColpVC, Col156 | 2013 | 6 |
| JAIHBZ000000000 | N18-00468 | **ST33** | Hadar | Raccoon | STR-TCY | aph(3')-Ib, aph(6)-Id, tet(A) | Col440 | 2013 | 6 |

[a] AMP = ampicillin; SOX = sulfisoxazole; STR = streptomycin; SXT = trimethoprim sulfamethoxazole; TCY = tetracycline.

Accession numbers of sequence data for all *Salmonella* isolates included in this study are available in S1 File. Most of these isolates were obtained from samples collected in 2012 (n = 82, 52%) and in 2011 (n = 50, 31%), with fewer isolates in 2013 (n = 27, 17%). Isolates originated from 80 unique raccoons; among animals captured multiple times, eight individuals contributed two isolates, and two raccoons contributed three isolates from different trapping dates. The majority of *Salmonella* isolates were phenotypically pan-susceptible to the antimicrobials tested: 96.7% of raccoon isolates (95%CI: 90.8–99.3%), 95.6% of soil isolates (95%CI: 85.2–99.5%), and 95.3% of swine manure pit isolates (95%CI: 76.2–99.9%). Six of the 159 isolates demonstrated phenotypic resistance (Table 1), and the overall prevalence of multidrug resistance (3+ drug classes) was 1.9% (n = 3/159, 95%CI: 0.4–5.4%). These multidrug resistant isolates were identified in two raccoon samples and one swine manure pit sample (Table 1).

**E. coli.** A total of 96 phenotypically resistant *E. coli* isolates were included, with the following source distribution: raccoon (n = 20), soil (n = 45), and swine manure pit (n = 31). Accession numbers for sequence data from all *E. coli* isolates can be found in S1 File. Most of these isolates were obtained from samples collected in 2013 (n = 39, 41%), followed by 2011 (n = 37, 39%), and 2012 (n = 20, 21%). Phenotypically resistant raccoon isolates were obtained from 20 unique individuals, with no repeated sampling. Overall, 26.0% of these resistant isolates were multidrug resistant (3+ drug classes) based on phenotype (n = 25/96), with most of these isolates identified in soil samples (n = 13), followed by raccoon samples (n = 7), and swine manure pit samples (n = 5). The corresponding prevalence of multidrug resistance was highest among resistant raccoon isolates (35.0%, 95%CI: 15.4–59.2%) and resistant soil isolates (28.9%, 95%CI: 16.4–44.3%), and lowest in resistant swine manure pit isolates (16.1%, 95%CI: 5.4–33.7%).

### Distribution of serovars and MLST types

**Salmonella.** In total, 21 sequence types representing 21 different serovars were identified (Fig 1 and Table 2). Three isolates were not typeable by MLST (two raccoon, one soil) due to a missing allele, or a partial match. The four most common serovars identified among all source types, in descending order, were *S*. Newport (28%), *S*. Agona (18%), *S*. Infantis (11%), and *S*. Typhimurium (9%; Fig 1). A number of internationally-recognized sequence types [27] were also identified; seventeen *S*. Infantis ST32 isolates were identified on all farms from all source

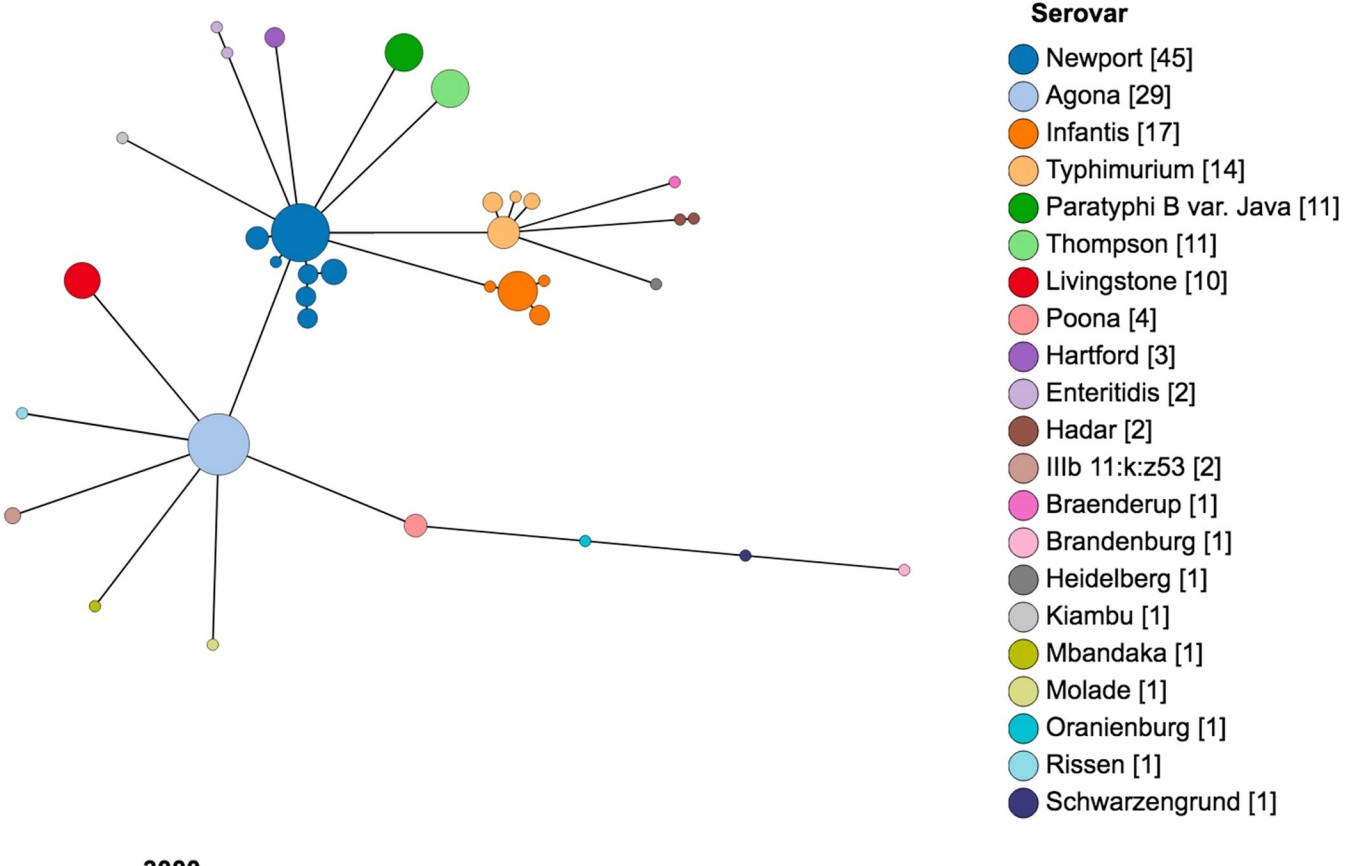

**Fig 1. Population structure of 159 *Salmonella enterica* isolates from raccoons, swine manure pits, and soil samples on swine farms in southern Ontario, Canada based on 3002-loci cgMLST scheme from Enterobase.** Minimum spanning tree created using $k = 30$ clustering threshold in *GrapeTree*. Serovars determined using SISTR. Three isolates (2 raccoon, 1 soil) were not typeable by MLST. Frequency counts are in square brackets. Bubble size is proportional to the number of isolates in each cluster, and each cluster contains isolates differing at a maximum of 30 cgMLST loci.

types (i.e., manure, raccoon and soil), and, apart from one sample, all were collected in 2011 and 2012. A total of fourteen *S.* Typhimurium ST19 isolates were identified in all sources on four of five farms; nine of these (64%) were obtained from the same farm (farm 6) in 2012. Finally, one isolate each of *S.* Schwarzengrund ST96, *S.* Heidelberg ST15, and *S.* Brandenburg ST65 were isolated from two raccoons and one swine manure pit sample, respectively.

**E. coli.** This population of resistant *E. coli* was comprised of 49 sequence types, of which two isolates from manure pit samples were not typeable by MLST (S1 Table), due to a missing allele or a partial match. None of the serovars identified here overlapped with those responsible for the majority of Shiga-toxin producing *E. coli* infections in humans (i.e., O157, O26, O45, O103, O111, O121, O145; S2 Table) [28]. Apart from eleven ST10 isolates (11.5%), no other major sequence types associated with uropathogenic *E. coli* (UPEC) strains in humans were identified (e.g., ST131, ST96, ST73, ST127, ST140) [29].

## Population structure based on cgMLST

**Salmonella.** The following *Salmonella* serovars were identified in both swine manure pit and raccoon isolates: *S.* Agona, *S.* Infantis, *S.* Poona, *S.* Typhimurium (Fig 2). Identical or similar cgMLST subtypes were identified from all sources for both *S.* Agona and *S.* Poona serovars.

**Table 2. Frequency of *Salmonella enterica* legacy multi-locus sequence types of isolates obtained from raccoons, swine manure pits, and soil samples on swine farms in southern Ontario, Canada 2011–2013 (n = 159[a]).**

| Sequence type[b] (Serovar) | Source type | | | Total[c] (%) |
|---|---|---|---|---|
| | Raccoon (n = 92) | Swine manure pit (n = 21) | Soil (n = 46) | |
| ST350 (Newport) | 31 | 0 | 11 | 42 (26.4%) |
| ST13 (Agona) | 16 | 5 | 8 | 29 (18.2%) |
| **ST32 (Infantis)** | 9 | 3 | 5 | 17 (10.7%) |
| **ST19 (Typhimurium)** | 7 | 1 | 6 | 14 (8.8%) |
| ST26 (Thompson) | 6 | 0 | 5 | 11 (6.9%) |
| ST404 (Paratyphi B var. Java) | 9 | 0 | 2 | 11 (6.9%) |
| ST638 (Livingstone) | 0 | 9 | 1 | 10 (6.3%) |
| **ST15 (Heidelburg)** | 1 | 0 | 0 | 1 (0.6%) |
| **ST96 (Schwarzengrund)** | 1 | 0 | 0 | 1 (0.6%) |
| **ST65 (Brandenburg)** | 0 | 1 | 0 | 1 (0.6%) |

Bolded STs represent internationally recognized sequence types implicated in human illness.

[a] Three isolates (two from raccoons, one from soil) were not typeable.

[b] Sequence types (STs) determined using 7-loci Achtman scheme.

[c] Other STs identified within 5 or fewer isolates were: ST413 (Mbandaka; n = 1), ST2848 (IIIb 11:k:z53; n = 2), ST22 (Braenderup; n = 1), ST23 (Oranienburg; n = 4), ST33 (Hadar; n = 2), ST309 (Kiambu; n = 1), ST308 (Poona; n = 4), ST405 (Hartford; n = 3), ST469 (Rissen; n = 1), ST11 (Enteritidis; n = 2), ST544 (Molade; n = 1).

The 29 *S.* Agona isolates had between 0 and 15 allelic differences; these isolates were identified in 2011 and 2012 on three of five farms. Four *S.* Poona isolates that differed by a maximum of 3 loci were isolated on farm 8 in 2013 from all sources. *Salmonella* Infantis (n = 17) and *S.* Typhimurium (n = 14) isolates clustered into single groups at thresholds of 51 and 329 allelic differences, respectively. Less commonly identified serovars that were isolated from both raccoon and soil isolates differed by a variety of minimum and maximum allelic differences between the two sources: *S.* Hadar (39 allelic differences; n = 2), *S.* Enteritidis (270 allelic differences; n = 2), *S.* Hartford (1–15 allelic differences; n = 3), *S.* Thompson (0–36 allelic differences; n = 11), *S.* Paratyphi B var. Java (0–28 allelic differences; n = 11), and *S.* Newport (0–68 allelic differences; n = 45). *Salmonella* Hadar and *S.* Hartford isolates were isolated from the same farm in different months of the same year. The majority of *S.* Thompson isolates were obtained from farm 8 (n = 8/10), and most were collected in 2012 (n = 7/10).

***E. coli.*** The population structure of *E. coli* based on cgMLST by source type and by farm is presented in Fig 3A and 3B; similar or identical subtypes were identified in isolates from raccoons, soil, and swine manure pit samples, regardless of farm location.

## In silico determination of acquired AMR genes and plasmid Inc types

**Salmonella.** Eighteen different Inc types and nine different AMR genes were identified in this *Salmonella* population (Tables 1 and 3). AMR genes identified were *aadA2*, *aadA4*, *aph (3")-Ib*, *aph(6)-Id*, *fosA7*, *sul1*, *tet(A)*, *tet(B)*, and *bla*$_{TEM-1}$. Gene *fosA7* was only identified in phenotypically pan-susceptible isolates, with an overall prevalence of 19.5% (n = 31/159; note that fosfomycin was not included on our antimicrobial test panel). All six phenotypically resistant *Salmonella* isolates were isolated in 2013 (Table 1). An isolate from a manure pit and an isolate from a raccoon with the same phenotypic resistance patterns (SOX-STR-TCY) previously identified as *S.* Typhimurium ST19 DT104 by Bondo et al., [19] contained the same resistance genes, and some of the same predicted plasmids. These two *S.* Typhimurium ST19 DT104 isolates differed from each other at 11 cgMLST loci, and by 28 SNPs; both were isolated

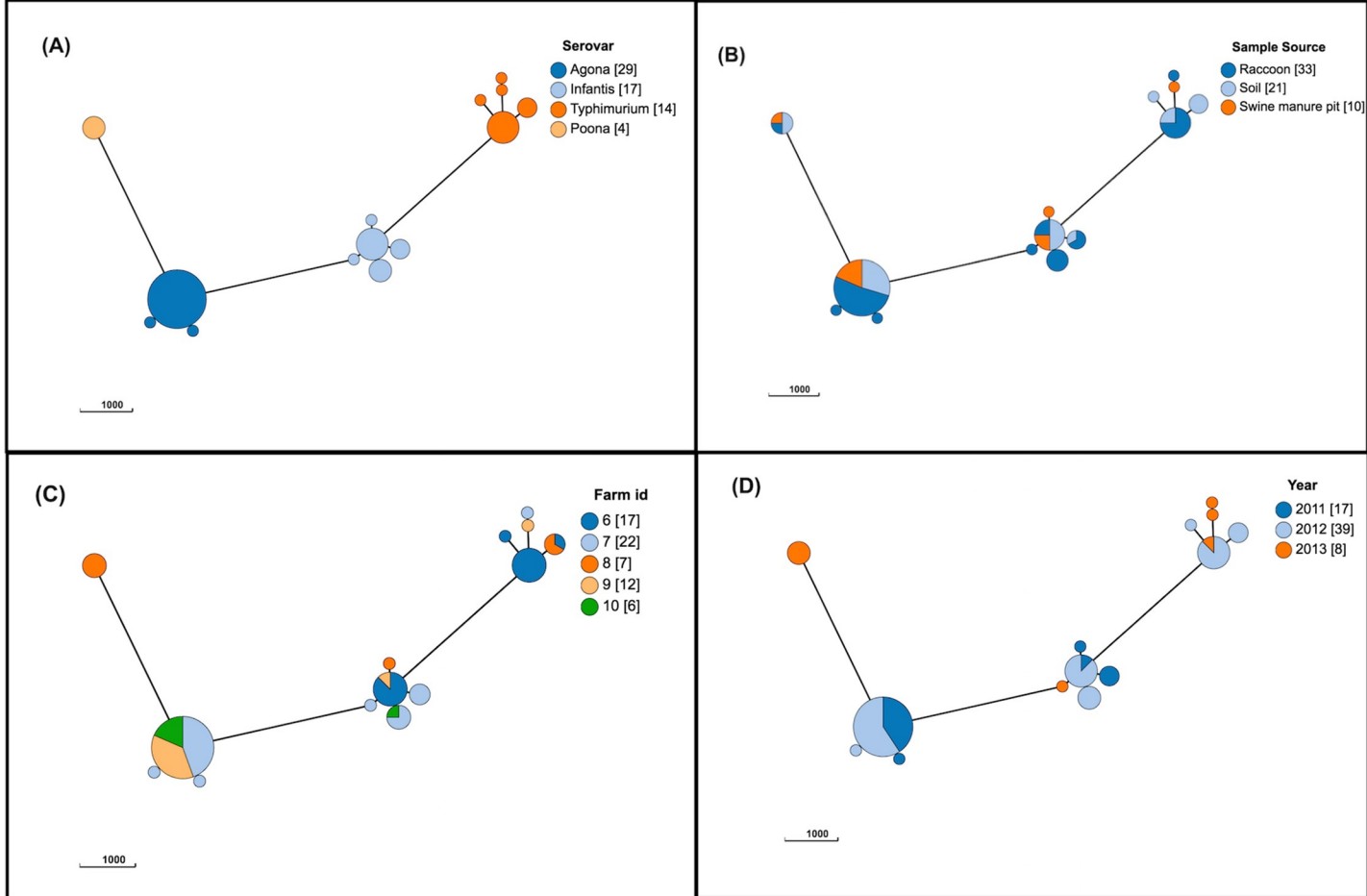

**Fig 2. Population structure of 64 isolates of *Salmonella enterica* isolates from raccoons, swine manure pits, and soil on swine farms in southern Ontario, Canada based on 3002-loci cgMLST scheme from Enterobase, for serovars *S*. Agona, *S*. Infantis, *S*. Typhimurium, and *S*. Poona (only serovars identified both in raccoon and swine manure pit samples).** Minimum spanning tree created using *k* = 5 clustering threshold in *GrapeTree*. (A) Population structure with serovars determined using SISTR. (B) Distribution by source type. (C) Distribution by farm. (D) Distribution by year of sampling. Frequency counts are in square brackets. Bubble size is proportional to the number of isolates in each cluster, and each cluster contains isolates differing at a maximum of 5 cgMLST loci.

in July 2013, but they were collected on different farms within 3km of one another. Two *S*. Hadar ST33 isolates, one from a raccoon and another from soil displayed the same phenotypic resistance pattern (STR-TCY) mediated by the same resistance genes (*tet[A], aph[6]-Id),* but carried different Inc types. These two *S*. Hadar ST33 isolates differed at 39 cgMLST loci and by 108 SNPs, and both were isolated in different months from the same farm.

**E. coli.** A total of 27 resistance genes and 21 Inc types were identified among resistant *E. coli* isolates (Tables 3 and 4). The distribution of resistance genes among different sources is presented in Table 4. The majority of genes identified confer resistance to aminoglycosides, tetracyclines, and folate pathway inhibitors. Genes conferring resistance to phenicols were uncommonly identified, and no macrolide resistance genes were identified (Table 4). Besides $bla_{\text{TEM-1}}$ (26.0% prevalence), only one other type of beta-lactamase resistance conferring gene, $bla_{\text{CMY-2}}$, was identified, and occurred in a single *E. coli* O9:H9 ST10 isolate collected from a raccoon (Table 4). This isolate containing the sole $bla_{\text{CMY-2}}$ also displayed phenotypic resistance to five of seven drug classes examined (AMC-AMP-FOX-TIO-CRO-CHL-STR-SOX-T-CY-SXT), and contained genes *aadA2*, *sul2* and *dfrA12*, as well as a single Inc type (IncC).

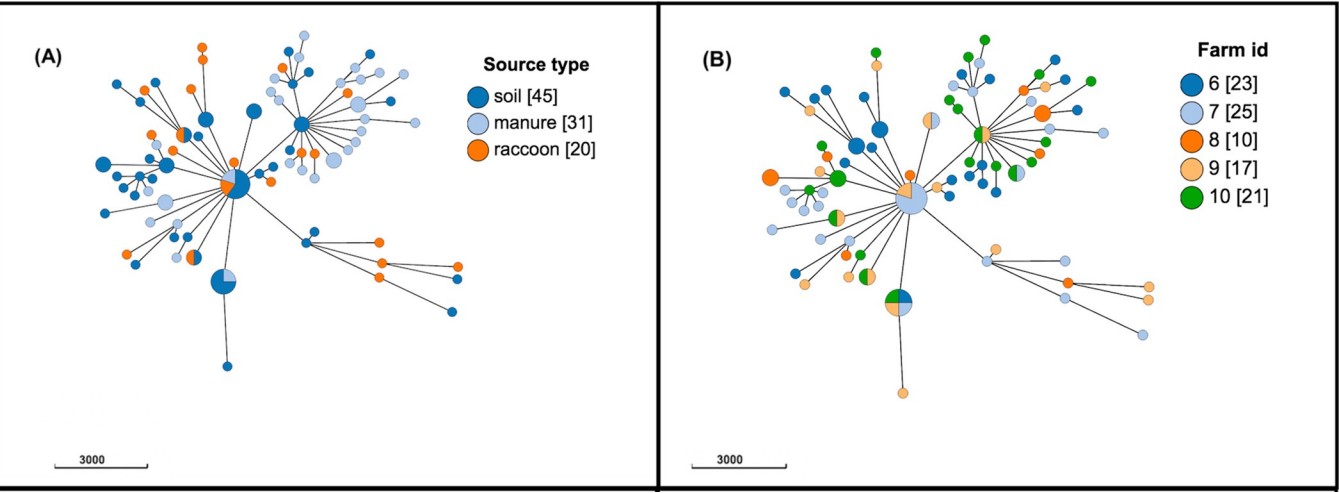

**Fig 3. Population structure of 96 phenotypically resistant *Escherichia coli* isolates from raccoons, swine manure pits, and soil on swine farms in southern Ontario, Canada based on 2513-loci cgMLST scheme from Enterobase.** Minimum spanning tree created using *k = 50* clustering threshold in *GrapeTree*. (A) Distribution by source type. (B) Distribution by farm location. Frequency counts are in square brackets. Bubble size is proportional to the number of isolates in each cluster, and each cluster contains isolates differing at a maximum of 50 cgMLST loci.

**Table 3. Frequencies of plasmid incompatibility (Inc) types identified using whole-genome sequencing data from *Salmonella enterica* and phenotypically resistant *Escherichia coli* isolates obtained from raccoons, swine manure pits, and soil samples on swine farms in southern Ontario, Canada 2011–2013.**

| | *Salmonella*[a] (n = 159) | *E. coli*[b] (n = 96) |
|---|---|---|
| **Rep-type** | **Count (%)** | **Count (%)** |
| IncFIB(AP001918) | 0 (0%) | 41 (42.7%) |
| IncI1(alpha) | 0 (0%) | 12 (12.5%) |
| IncFiip96a | 59 (37.1%) | 0 (0%) |
| IncFII | 0 (0%) | 15 (15.6%) |
| IncFIIS | 64 (40.2%) | 0 (0%) |
| IncFIA | 0 (0%) | 9 (9.4%) |
| IncY | 3 (1.9%) | 11 (11.5%) |
| IncX1 | 27 (17.0%) | 7 (7.3%) |
| IncX3 | 25 (15.7%) | 0 (0%) |
| IncQ1 | 0 (0%) | 5 (5.2%) |
| IncI1(I-gamma) | 5 (3.1%) | 0 (0%) |
| IncR | 0 (0%) | 8 (8.3%) |
| ColYe4449 | 28 (17.6%) | 0 (0%) |
| ColRNAI | 6 (3.8%) | 0 (0%) |
| ColpHAD28 | 7 (4.4%) | 0 (0%) |
| p0111 | 0 (0%) | 8 (8.3%) |

[a] Plasmid Inc types identified in fewer than five *Salmonella* isolates included: ColpVC (n = 2), IncFIBS (n = 5), Col440ii (n = 3), IncHI2 (n = 1), pkpccav1321 (n = 1), IncHI2A (n = 1), Col156 (n = 3), Col440i (n = 2), IncFIBphcm2 (n = 1).

[b] Plasmid Inc types identified in fewer than five *E. coli* isolates included: ColBS512 (n = 2), ColE10 (n = 2), ColpVC (n = 1), IncC (n = 2), IncB/O/K/Z (n = 1), IncFIA(HI1) (n = 3), IncFIB(K) (n = 2), IncFIB(pB171) (n = 1), IncFIC (FII) (n = 3), IncFII(pHN7A8) (n = 1), IncHI2A (n = 2), IncHI2 (n = 2).

**Table 4. Frequencies of acquired antimicrobial resistance genes identified using whole-genome sequencing data from phenotypically resistant *Escherichia coli* isolates obtained from raccoons, swine manure pits, and soil samples on swine farms in southern Ontario, Canada 2011–2013 (n = 96).**

| Antimicrobial group | Resistance gene | Accession no.[†] | Raccoon(n = 20) | Swine manure pit (n = 31) | Soil (n = 45) | Total (%) |
|---|---|---|---|---|---|---|
| Aminoglycoside | *aac(3)-IVa* | NC_009838 | 1 | 0 | 0 | 1 (1.0%) |
| | *aadA2* | JQ364967 | 3 | 1 | 1 | 5 (4.8%) |
| | *ant(3")-Ia* | X02340 | 4 | 7 | 8 | 19 (19.8%) |
| | *aph(3')-Ia* | V00359/EF015636 | 1 | 0 | 2 | 3 (3.1%) |
| | *aph(3')-IIa* | V00618 | 0 | 1 | 0 | 1 (1.0%) |
| | *aph(3")-Ib* | AF321551/AF024602 | 11 | 13 | 20 | 44 (45.8%) |
| | *aph(6)-Ic* | X01702 | 0 | 1 | 0 | 1 (1.0%) |
| | *aph(6)-Id* | M28829 | 11 | 13 | 20 | 44 (45.8%) |
| Beta-lactam | *bla*CMY-2 | X91840 | 1 | 0 | 0 | 1 (1.0%) |
| | *bla*TEM-1 | AY458016/HM749966/FJ560503 | 7 | 7 | 11 | 25 (26.0%) |
| Lincosamide | *lnuC* | AY928180 | 0 | 0 | 1 | 1 (1.0%) |
| | *lnuF* | EU118119 | 0 | 0 | 1 | 1 (1.0%) |
| Folate pathway inhibitors | *dfrA1* | AF203818/X00926 | 1 | 1 | 4 | 6 (6.2%) |
| | *dfrA5* | X12868 | 0 | 1 | 3 | 4 (4.2%) |
| | *drfA12* | AM040708 | 2 | 0 | 0 | 2 (2.0%) |
| | *dfrA14* | DQ388123 | 2 | 0 | 1 | 3 (3.1%) |
| | *dfrA23* | AJ746361 | 1 | 0 | 0 | 1 (1.0%) |
| | *sul1* | EU780013 | 5 | 2 | 4 | 11 (11.5%) |
| | *sul2* | HQ840942/AY034138 | 6 | 1 | 9 | 16 (16.7%) |
| | *sul3* | AJ459418 | 1 | 1 | 2 | 4 (4.2%) |
| Phenicol | *floR* | AF118107 | 3 | 1 | 3 | 7 (7.3%) |
| | *cmlA1* | M64556 | 1 | 1 | 1 | 3 (3.1%) |
| Fosfomycin | *fosA7* | LAPJ01000014 | 1 | 0 | 2 | 3 (3.1%) |
| Tetracycline | *tet(A)* | AF534183 | 9 | 11 | 25 | 45 (46.9%) |
| | *tet(B)* | AF326777/AP000342 | 5 | 16 | 15 | 36 (37.5%) |
| | *tet(C)* | AY046276/AF055345 | 0 | 0 | 2 | 2 (2.1%) |
| | *tet(M)* | X04388 | 0 | 0 | 1 | 1 (1.0%) |

[†] Values from *Resfinder* database.

Two additional isolates were phenotypically resistant to five of the seven drug classes examined: a soil isolate (AMP-CHL-KAN-STR-SOX-TCY-SXT), and a swine manure pit isolate (AMP-CHL-STR-SOX-TCY). Despite having similar phenotypic resistance patterns, these isolates represented different sequence types (ST106 [soil] and ST542 [manure pit]), contained different predicted plasmids, and, apart from the presence of *tet(A)* and *bla*TEM-1, contained a different profile of genes responsible for conferring resistance.

## Comparison of *Salmonella* and *E. coli*

Among Inc types commonly identified in this population of *Salmonella* and *E. coli*, few incompatibility types were identified in both organisms (Table 3). The majority of Inc types were restricted to either *Salmonella* or *E. coli*, but not found in both (e.g., IncFIB[AP001918] in *E. coli*, IncFiip96a in *Salmonella*). We also evaluated whether resistance genes may be shared between *E. coli* and *Salmonella* isolates within the same animal; of the three raccoon samples positive for resistant *Salmonella* (Table 1), no corresponding resistant *E. coli* were isolated from the same animal during the study period, either on the same capture date or on another capture date. Along with a single resistant *Salmonella* from a swine manure pit sample

**Table 5. Test sensitivity and specificity[a] for *in silico* identification of acquired antimicrobial resistance genes in *Salmonella enterica* and phenotypically resistant *Escherichia coli* isolates obtained from raccoons, swine manure pits, and soil samples on swine farms in southern Ontario, Canada 2011–2013.**

| Antimicrobial class | *Salmonella* (n = 159) | | *Escherichia coli* (n = 96) | |
|---|---|---|---|---|
| | Test sensitivity (95%CI) | Test specificity (95%CI) | Test sensitivity (95%CI) | Test specificity (95%CI) |
| Aminoglycoside | 100% (39.8–100%*) | 99.4% (96.5–99.9%) | 100% (92.7–100*%) | 80.9% (66.7–90.9%) |
| Beta-lactam | 100% (2.5–100%*) | 100% (97.7–100%*) | 89.6% (72.7–97.8%) | 100% (94.6–100%*) |
| Macrolide | —[c] | 100% (97.7–100%*) | —[c] | 100% (96.2–100%*) |
| Sulfonamide | 100% (29.2–100%*) | 100% (97.7–100%*) | 100% (86.8–100%*) | 98.6% (92.3–99.9%) |
| Phenicol | —[c] | 100% (97.7–100%*) | 100% (66.4–100%*) | 100% (95.8–100%*) |
| Tetracycline | 100% (54.1–100%*) | 100% (97.6–100%*) | 100% (95.5–100%*) | 93.3% (68.1–99.8) |
| **Overall[b]** | **100% (76.8–100%*)** | **99.9% (99.4–99.9%)** | **98.5% (95.5–99.7%)** | **97.1% (94.9–98.6%)** |

[a] Phenotypic antimicrobial resistance test results were considered the gold standard. Detection of 15 antimicrobials performed using the CMV3AGNF panel from National Antimicrobial Resistance Monitoring System (Sensititre, Thermo Scientific). *In silico* acquired resistance genes detected using Abricate and the Resfinder database.

[b] Raw counts for all isolates and antimicrobials were pooled together.

[c] Not applicable since no phenotypic resistance was identified.

* One-sided, 97.5% confidence interval.

originating from farm 9, three resistant *E. coli* from manure pits were obtained from the same farm, with one collected in the same year; however, apart from two genes in common between two of the *E. coli* isolates and the *Salmonella* isolate (i.e., *sul1*, *tet[A]*), there was no overlap with regards to resistance genes, resistance patterns, or Inc types between these resistant *E. coli* isolates and the resistant *Salmonella* isolate.

## Sensitivity and specificity of in silico AMR prediction

Detailed results from phenotypic antimicrobial susceptibility testing were previously reported by Bondo et al. [18, 19]. Test sensitivity could not be assessed for certain drug classes where no phenotypic resistance was identified (e.g., macrolides, phenicols; Table 5). Test sensitivity and specificity were 89% or greater for all drug classes in both *Salmonella* and *E. coli*, with the exception of test specificity in *E. coli* for aminoglycosides (80.9%). The overall test sensitivity and specificity (i.e., all raw counts pooled together) was 97% or greater for both organisms.

## Statistical results

**Salmonella.** All five Inc types and the one resistance gene analyzed (*fosA7*) were significantly associated with at least one independent variable (Table 6). Among the four predicted plasmids associated with source type (i.e., IncX1, IncFIIS, IncX3, IncFiip96a), the odds of identifying these Inc types were consistently greater in raccoons compared to swine manure pit isolates, and in some cases, the odds were also greater in soil isolates compared to swine manure pits (i.e., IncFIIS, IncFiip96a; Table 6). Two Inc types (i.e., IncX3, Colye4449) and *fosA7* were significantly associated with farm location. Contrasts concerning farm location are available in S3 Table. Colye4449 was the only outcome associated with year of sampling and its prevalence was significantly lower in 2013 compared to 2011 and 2012 (Table 6). The random intercept was not retained in any model with a statistically significant fixed effect, since it did not improve the fit of the model, the variance component was negligible ($<1 \times 10^{-3}$), and/or coefficients could not be estimated without exact logistic regression.

**E. coli.** Of the eight Inc types and four resistance genes examined statistically, only two Inc types (i.e., IncI1[alpha], IncFIB[AP001918]) and one gene (*sul2*) were significantly

**Table 6. Univariable logistic regression models[a,b,c] assessing the association between source type, farm location, and year of sampling and the occurrence of select antimicrobial resistance genes and plasmid incompatibility (Inc) types in *Salmonella enterica* isolates from raccoons, swine manure pits, and soil samples on swine farms in southern Ontario, Canada 2011–2013 (n = 159).**

| | | IncX1 | | IncFIIS | | IncX3 | |
|---|---|---|---|---|---|---|---|
| | | OR (95%CI) | *p*-value | OR (95%CI) | *p*-value | OR (95%CI) | *p*-value |
| **Source type** | Swine manure pit | REF | 0.012 (global)[a] | REF | <0.001 (global)[b] | REF | 0.012 (global)[a] |
| | Raccoon | 9.06* (1.47–∞) | 0.012 | 20.00 (2.57–155.28) | 0.004 | 8.52* (1.37–∞) | 0.025 |
| | Soil | 3.30* (0.42–∞) | 0.173 | 11.72 (1.44–95.33) | 0.021 | 2.54* (0.30–∞) | 0.300 |
| **Farm** | 6 | REF | 0.288 (global)[c] | REF | 0.089 (global)[c] | REF | <0.001 (global)[a] |
| | 7 | 6.33 (0.45–99.89) | 0.171 | 30.27 (1.63–562.91) | 0.022 | 9.36* (1.39–∞) | 0.011 |
| | 8 | 0.73 (0.05–9.83) | 0.811 | 71.63 (2.98–1721.31) | 0.008 | 2.12* (0.22–∞) | 0.310 |
| | 9 | 0.71 (0.02–22.64) | 0.849 | 7.65 (0.36–163.68) | 0.193 | 1.41* (0.04–∞) | 0.415 |
| | 10 | 25.34 (0.80–797.52) | 0.066 | 242.30 (4.50–13042.56) | 0.007 | 18.9* (2.80–∞) | 0.001 |
| **Year** | 2011 | REF | 0.182 (global)[b] | REF | 0.420 (global)[c] | REF | 0.230 (global)[c] |
| | 2012 | 2.53 (0.87–7.32) | 0.086 | 0.46 (0.14–1.48) | 0.190 | 5.72 (0.55–59.64) | 0.145 |
| | 2013 | 1.56 (0.38–6.39) | 0.533 | 0.69 (0.15–3.19) | 0.641 | 0.66 (0.05–9.13) | 0.755 |
| | | IncFiip96a | | Colye4449[d] | | *fosA7* | |
| | | OR (95%CI) | *p*-value | OR (95%CI) | *p*-value | OR (95%CI) | *p*-value |
| **Source type** | Swine manure pit | REF | <0.001 (global)[a] | REF | 0.758 (global)[c] | REF | 0.831 (global)[b] |
| | Raccoon | 26.39* (4.37–∞) | <0.001 | 0.62 (0.17–2.19) | 0.457 | 0.78 (0.25–2.40) | 0.663 |
| | Soil | 13.54 (2.10–∞) | 0.003 | 0.68 (0.17–2.67) | 0.580 | 0.67 (0.19–2.38) | 0.539 |
| **Farm** | 6 | REF | 0.117 (global)[c] | REF | <0.001 (global)[a] | REF | <0.001 (global)[b] |
| | 7 | 95.69 (1.58–5800.67) | 0.029 | 13.40* (2.06–∞) | 0.003 | 10.73 (1.31–87.67) | 0.027 |
| | 8 | 325.49 (3.35–31575.21) | 0.013 | 1.00 (0–∞) | NE | 0.52 (0.03–8.75) | 0.652 |
| | 9 | 20.59 (0.31–1363.38) | 0.157 | 40.00* (5.54–∞) | <0.001 | 32.86 (3.56–303.41) | 0.002 |
| | 10 | 1735.86 (6.29–478596) | 0.009 | 6.32* (0.81–∞) | 0.056 | 4.79 (0.52–44.20) | 0.167 |
| **Year** | 2011 | REF | 0.319 (global)[c] | REF | 0.026 (global)[a] | REF | 0.241 (global)[c] |
| | 2012 | 0.41 (0.10–1.65) | 0.209 | 0.77 (0.30–1.98) | 0.661 | 0.81 (0.33–2.00) | 0.644 |
| | 2013 | 0.25 (0.03–1.82) | 0.173 | 0.09* (0–0.57) | 0.006 | 0.23 (0.04–1.27) | 0.093 |

REF = referent group, CI = confidence interval, NE = not estimated.

* Median unbiased estimates obtained with exact logistic regression.

[a] Exact logistic regression model.

[b] Ordinary logistic regression model.

[c] Multi-level model. A random intercept to account for repeated sampling of animals and swine manure pits was retained: IncX1 farm intraclass correlation coefficient (ICC): 55.7% (95%CI: 8.1–94.7%); IncFIIS farm ICC: 61.9% (95%CI: 24.8–88.9%); IncFIIS year ICC: 53.6% (95%CI: 19.2–84.9%); IncX3 year ICC: 54.4% (95%CI: 18.2–86.5%); IncFiip96a farm ICC: 70.1% (95%CI: 32.6–91.9%); IncFiip96a year ICC: 64.5% (95%CI: 29.4–88.8%); Colye4449 source ICC: 6.8% (95%CI: 0.0–99.2%); *fosA7* year ICC: 10.3% (95%CI: 0.0–96.1%).

[d] The odds of Colye4449 were significantly lower in 2013 compared to 2012 (OR: 0.11*, 95%CI: 0–0.70).

** Contrasts are available in S3 Table.

associated with source or farm location, and none were associated with the year of sampling (Table 7). IncI1(alpha) was significantly associated with farm location, whereas *sul2* and IncFIB(AP001918) were associated with source type, and both were detected more frequently in raccoons compared to swine manure isolates. Contrasts are available in S4 Table. Although model assumptions were met for *sul2* and IncFIB (AP001918) models, the random intercept was not retained since the variance components were negligible ($<1\times10^{-3}$), and the BIC favoured models without the random intercept.

**Table 7. Univariable logistic regression models[a,b,c] assessing the association between source type, farm location, and year of sampling, and the occurrence of select antimicrobial resistance genes and plasmid incompatibility (Inc) types in phenotypically resistant *Escherichia coli* isolates obtained from raccoons, swine manure pits, and soil samples on swine farms in southern Ontario, Canada 2011–2013 (n = 96).**

| | | *tet(A)* | | *tet(B)* | | *bla*TEM-1 | | *sul1* | |
|---|---|---|---|---|---|---|---|---|---|
| | | OR (95% CI) | *p*-value | OR (95% CI) | *p*-value | OR (95% CI) | *p*-value | OR (95% CI) | *p*-value |
| **Source type** | Swine manure pit | REF | **0.220 (global)[a]** | REF | **0.117 (global)[a]** | REF | **0.594 (global)[a]** | REF | **0.133 (global)[a]** |
| | Raccoon | 1.49 (0.47–4.69) | 0.498 | 0.47 (0.18–1.20) | 0.113 | 1.85 (0.53–6.41) | 0.335 | 4.83 (0.84–27.93) | 0.078 |
| | Soil | 2.27 (0.89–5.83) | 0.088 | 0.31 (0.09–1.07) | 0.064 | 1.11 (0.38–3.27) | 0.851 | 1.41 (0.24–8.24) | 0.700 |
| **Farm** | 6 | REF | **0.798 (global)[a]** | REF | **0.165 (global)[a]** | REF | **0.060 (global)[a]** | REF | **0.676 (global)[a]** |
| | 7 | 1.98 (0.63–6.26) | 0.245 | 0.41 (0.12–1.41) | 0.157 | 0.49 (0.10–2.34) | 0.372 | 0.58 (0.04–5.66) | 0.660 |
| | 8 | 1.56 (0.35–6.94) | 0.563 | 1.30 (0.29–5.76) | 0.730 | 0.90 (0.14–5.66) | 0.911 | 0.56* (0–5.65) | 0.536 |
| | 9 | 1.09 (0.30–3.91) | 0.896 | 0.40 (0.10–1.61) | 0.197 | 4.05 (1.02–16.00) | 0.046 | 1.41 (0.16–12.20) | 0.999 |
| | 10 | 1.41 (0.43–4.68) | 0.571 | 1.43 (0.43–4.69) | 0.555 | 1.44 (0.36–5.67) | 0.602 | 1.11 (0.13–9.37) | 0.999 |
| **Year** | 2011 | REF | **0.077 (global)[a]** | REF | **0.417 (global)[a]** | REF | **0.836 (global)[a]** | REF | **0.688 (global)[a]** |
| | 2012 | 0.32 (0.09–1.05) | 0.060 | 2.08 (0.68–6.35) | 0.197 | 1.16 (0.35–3.84) | 0.812 | 1.99 (0.36–10.98) | 0.425 |
| | 2013 | 1.10 (0.45–2.72) | 0.828 | 1.17 (0.45–3.02) | 0.750 | 0.81 (0.29–2.29) | 0.691 | 1.67 (0.37–7.53) | 0.507 |

| | | *sul2* | | *ant(3")-Ia* | | *aph(3")-Ib* | | *aph(6)-Id* | |
|---|---|---|---|---|---|---|---|---|---|
| | | OR (95% CI) | *p*-value | OR (95% CI) | *p*-value | OR (95% CI) | *p*-value | OR (95% CI) | *p*-value |
| **Source type** | Swine manure pit | REF | **0.017 (global)[a]** | REF | **0.876 (global)[a]** | REF | **0.638 (global)[a]** | REF | **0.638 (global)[a]** |
| | Raccoon | 12.86 (1.41–117.20) | 0.024 | 0.86 (0.21–3.41) | 0.827 | 1.69 (0.54–5.26) | 0.363 | 1.69 (0.54–5.26) | 0.363 |
| | Soil | 7.5 (0.90–62.61) | 0.063 | 0.74 (0.24–2.31) | 0.606 | 1.11 (0.44–2.79) | 0.828 | 1.11 (0.44–2.79) | 0.828 |
| **Farm** | 6 | REF | **0.636 (global)[b]** | REF | **0.987 (global)[a]** | REF | **0.283 (global)[a]** | REF | **0.283 (global)[a]** |
| | 7 | 0.48 (0.02–13.29) | 0.666 | 1.19 (0.28–5.10) | 0.817 | 0.61 (0.19–1.92) | 0.399 | 0.61 (0.19–1.92) | 0.399 |
| | 8 | 0.22 (0.00–31.73) | 0.555 | 1.19 (0.18–7.84) | 0.858 | 2.14 (0.44–10.39) | 0.346 | 2.14 (0.44–10.39) | 0.346 |
| | 9 | 0.05 (0.00–4.42) | 0.190 | 1.02 (0.19–5.29) | 0.983 | 0.38 (0.10–1.44) | 0.155 | 0.38 (0.10–1.44) | 0.155 |
| | 10 | 2.97 (0.10–90.37) | 0.532 | 1.48 (0.34–6.48) | 0.599 | 0.83 (0.25–2.72) | 0.763 | 0.83 (0.25–2.72) | 0.763 |
| **Year** | 2011 | REF | **0.788 (global)[a]** | REF | **0.922 (global)[a]** | REF | **0.354 (global)[a]** | REF | **0.354 (global)[a]** |
| | 2012 | 1.60 (0.38–6.79) | 0.524 | 0.91 (0.23–3.48) | 0.886 | 1.97 (0.65–5.95) | 0.230 | 1.97 (0.65–5.95) | 0.230 |
| | 2013 | 1.40 (0.40–4.87) | 0.597 | 0.79 (0.25–2.46) | 0.688 | 0.91 (0.37–2.27) | 0.845 | 0.91 (0.37–2.27) | 0.845 |

| | | IncFIB(AP001918) | | IncI1(alpha) | | IncFII | | IncY | |
|---|---|---|---|---|---|---|---|---|---|
| | | OR (95% CI) | *p*-value | OR (95% CI) | *p*-value | OR (95% CI) | *p*-value | OR (95% CI) | *p*-value |

(*Continued*)

**Table 7.** (Continued)

| Source type | | | | | | | | | |
|---|---|---|---|---|---|---|---|---|---|
| Source type | Swine manure pit | REF | **0.013 (global)[a]** | REF | **0.907 (global)[a]** | REF | **0.196 (global)[a]** | REF | **0.998 (global)[b]** |
| | Wildlife | 5.14 (1.50–17.57) | 0.009 | 1.19 (0.24–5.99) | 0.832 | 0.18 (0.02–1.60) | 0.124 | 1.37 (1.8e-08–1.04e+08) | 0.973 |
| | Soil | 3.27 (1.18–9.14) | 0.023 | 0.84 (0.21–3.43) | 0.812 | 0.63 (0.20–2.03) | 0.440 | 1.14 (1.9e-08–6.5e+07) | 0.989 |
| Farm | 6 | REF | **0.807 (global)[b]** | REF | **0.028 (global)[c]** | REF | **0.343 (global)[a]** | REF | **0.050 (global)[c]** |
| | 7 | 1.95 (0.35–10.81) | 0.446 | 3.97 (0.64–43.91) | 0.140 | 4.19 (0.43–40.62) | 0.216 | 0.92 (0.01–75.19) | 0.999 |
| | 8 | 2.38 (0.22–25.77) | 0.476 | 0.93 (0.00–12.47) | 0.564 | 9.43 (0.84–105.79) | 0.069 | 2.3* (0.00–89.70) | 0.999 |
| | 9 | 1.26 (0.23–7.02) | 0.787 | 2.20 (0.22–29.58) | 0.634 | 4.71 (0.44–49.94) | 0.198 | 6.46 (0.56–348.13) | 0.144 |
| | 10 | 0.59 (0.09–3.69) | 0.574 | 0.44 (0–5.80) | 0.489 | 5.18 (0.53–50.65) | 0.158 | 6.60 (0.65–339.59) | 0.088 |
| Year | 2011 | REF | **0.512 (global)[a]** | REF | **0.227 (global)[b]** | REF | **0.865 (global)[a]** | REF | **0.518 (global)[b]** |
| | 2012 | 1.18 (0.39–3.50) | 0.770 | 0.54 (0.08–3.74) | 0.529 | 1.13 (0.24–5.31) | 0.878 | 0.07 (0.00–29.73) | 0.395 |
| | 2013 | 0.66 (0.26–1.65) | 0.374 | 0.13 (0.01–1.34) | 0.087 | 1.40 (0.40–4.88) | 0.597 | 0.23 (0.01–4.03) | 0.318 |

REF = referent group, OR = odds ratio.

* Median unbiased estimates obtained with exact logistic regression.

[a] Ordinary logistic model.

[b] Multi-level model. A random intercept to account for repeated sampling of animals and swine manure pits retained in the following models: *sul2* farm intraclass correlation coefficient (ICC): 74.3% (95%CI: 27.8–95.5%); IncFIB(AP001918) farm ICC: 32.1% (95%CI: 2.9–88.2%); IncI1(alpha) year ICC: 35.2% (95%CI: 2.4–92.1%); IncY source type ICC: 99.8% (95%CI: 98.9–99.9%); IncY year ICC: 99.8% (95%CI: 97.6–99.9%).

[c] Exact logistic regression model.

** The following contrasts were statistically significant (p<0.05): the odds of IncFIB(AP001918) were lower in swine compared to soil (OR: 0.30, 95%CI: 0.11–0.85); the odds of IncI1(alpha) were lower on farm 10 versus farm 7 (OR: 0.10*, 95%CI: 0–0.70).

## Discussion

Using a course-grained epidemiological approach informed by whole-genome sequence data to assess for potential transmission of *Salmonella*, *E. coli* and related AMR determinants at the source level provides evidence suggestive of local transmission of certain strains in swine farm environments, or exposure to a common environmental source. We frequently identified findings consistent with soil-raccoon transmission of *Salmonella*, *E. coli* and AMR determinants, but the evidence provided by our study suggests that there is limited transmission of *Salmonella* and associated resistance genes between raccoons and manure pits in the swine farm environment. *Salmonella* serovars with a broad-host affinity such as *S.* Typhimurium and *S.* Infantis [30] were identified in all sampling sources (i.e., raccoons, swine manure pits, soil); on average, these serovars displayed greater diversity in cgMLST profiles (>50 allelic differences) than serovars such as *S.* Poona and *S.* Agona, which were also identified in all sampling sources, and tended to be more related (differed at <15 cgMLST loci on average). Additionally, we identified "clusters" of certain *Salmonella* serovars that were found in multiple sources but restricted to certain years and farms (i.e., *S.* Poona on farm 8 in 2013; *S.* Typhimurium ST19 on farm 6 in 2012; *S.* Thompson on farm 8), albeit these observations are based on small sample sizes (n<20) and differences could not be tested statistically. All sources were found to

contain at least one or more internationally important *Salmonella* sequence types which have been implicated in human illness (e.g., *S.* Typhimurium ST19, *S.* Infantis ST32) [27], but, overall, very few isolates exhibited phenotypic AMR (<5%), as previously observed by Bondo et al. [19]. As a result of this low prevalence of AMR among *Salmonella*, and apart from one gene (*fosA7*), we were unable to examine patterns in the distribution of resistance genes or assess risk factors statistically. Conversely, our inclusion of untyped *E. coli* based on demonstrated phenotypic AMR unquestionably resulted in a sampling bias [31], but this approach enabled statistical assessments which contribute to a preliminary understanding of the dynamics and movement of AMR in enteric bacterial populations in these different sources.

Similar to other studies [32, 33], the use of *in silico* tools for the identification of resistance genes in this study was generally reliable, although we did not assess drug classes for which chromosomal resistance plays an important role (i.e., quinolones). The overall test sensitivity and specificity of genotypic AMR identification in our study (>97% for both *Salmonella* and *E. coli*) was comparable to, or greater than, values reported by these two other studies (range: 75–97%) [32, 33]; in some cases, these differences may be attributed–at least in part–to the antimicrobial panel and bioinformatics pipeline used, the population of *Salmonella* investigated (i.e., the sampling sources and serovars included), and the associated sample size (i.e., a small number of isolates resulting in wide confidence intervals around point estimates of sensitivity or specificity). The specificity for aminoglycoside resistance among *E. coli* in our study (80%) was lowest of all drug classes, which is also consistent with findings from both of these studies [32, 33], in which the specificity of genotypic identification of streptomycin using WGS data was also the lowest of all drugs evaluated by these studies. The presence of silent (i.e., unexpressed) resistance genes may account for these findings [34] and should be considered in future genotypic AMR evaluations which do not have access to corresponding phenotypic AMR data.

For the purposes of validating phenotypic results and investigating possible transmission of resistance genes between different sources, our *in silico* AMR data has provided insights about the movement of AMR determinants in a southern Ontario agroecosystem. In some cases, resistant *Salmonella* with the same phenotypic AMR patterns contained different genes responsible for conferring resistance. Our findings of similar cgMLST profiles (<10 allelic differences) [24], along with the presence of the same resistance genes among isolates that were spatially or temporally linked were suggestive of the dissemination of closely related isolates. Such cases included two *S.* Hadar ST33 (one soil, one raccoon) from the same farm in the same year, and two *S.* Typhimurium ST19 DT104 (one raccoon, one swine manure pit) from different farms in the same month and year; these findings highlight the potential occurrence of on-farm as well as between-farm transmission of *Salmonella* between different sources. Similar or identical cgMLST profiles were also frequently identified among raccoon and soil isolates for a variety of serovars, including, but not limited to, the following: *S.* Newport, *S.* Agona, *S.* Thompson, *S.* Hartford, *S.* Paratyphi B var. Java. In conjunction with previous work comparing samples from raccoon paws to their corresponding fecal samples [35], these particular findings suggest that transmission between raccoons and their immediate environment is likely occurring.

Among *E. coli* isolates, a total of 25 multidrug resistant isolates were identified. One such *E. coli* isolate identified in a raccoon with phenotypic AMR pattern AMC-AMP-FOX-TIO--CRO-CHL-STR-SOX-TCY-SXT contained the sole $bla_{CMY-2}$ identified within our study. In general, the types of genotypic resistance identified in these populations of *Salmonella* and *E. coli* conferred resistance to aminoglycosides, tetracyclines, and folate pathway inhibitors; other more concerning types of resistance to macrolides, phenicols, and fluoroquinolones were rarely identified, or absent altogether in this study.

Many predicted plasmids found in *Salmonella* were not associated with presence of resistance genes. Two Inc types were present in ~40% of all *Salmonella* isolates, the majority of which were pan-susceptible; the abundance of these particular *Salmonella* genotypes may be related to exposure to disinfectants or other environmental stressors at these sites, and related selection of plasmids containing virulence genes or genes conferring resistance to disinfectants or heavy metals (not evaluated here) [36]. The predicted plasmids appearing in *Salmonella* were very distinct from those identified in the resistant *E. coli* population; few Inc types were commonly identified in both organisms. Previous work by Varga et al. [37] demonstrated a lack of association between phenotypic resistance patterns for *Salmonella* and generic *E. coli* isolates originating from the same swine manure sample. We made a similar observation based on our examination of a small number of raccoons that carried both resistant *Salmonella* and resistant *E. coli*; we identified no similarities among AMR determinants between organisms originating from the same raccoon, manure pit, or soil sample. However, the examination of only one bacterial isolate per fecal sample may not capture or represent important and relevant aspects of the microbiome and related resistome [38].

Major differences in the distribution of AMR genes and predicted plasmids between *E. coli* and *Salmonella* were identified. The majority of AMR genes and Inc types in *E. coli* analyzed statistically were not significantly associated with any of the predictor variables, whereas all of the Inc types and the single gene analyzed in *Salmonella* (i.e., *fosA7*) were associated with source type or farm location, or both. The lack of association for most of the AMR determinants in *E. coli* with source, farm, or year of sampling suggests widespread sharing of *E. coli* AMR determinants between sources in this region, or common exposure to AMR pollution in environmental sources (e.g., water). However, these findings should be interpreted in light of our selection of a population of resistant *E. coli*, and future examination of both susceptible and resistant isolates will provide important context for these findings. For those Inc types and genes associated with farm location, the farm with the highest odds of these outcomes was not consistent and varied depending on the particular Inc type or gene under examination. In contrast, Inc types and genes associated with source type were consistently more likely to be identified in raccoons compared to swine manure pit isolates, and, for certain outcomes, they were also more likely to be isolated from soil compared to swine manure pits. These findings suggest that, for certain AMR determinants (particularly in *Salmonella*), limited exchange between raccoons and soil with swine manure pits is occurring, similar to findings from previous work in this study region examining *Campylobacter* isolates from raccoons and livestock (swine, dairy, beef) [39].

The identification of sequence types of international importance in raccoons, soil, and manure pits sampled in swine farm environments, in particular *S*. Typhimurium ST19 (the most prevalent sequence type among *S*. Typhimurium isolates globally [40]) is suggestive of widespread circulation of these strains in this region of Ontario. To date, there are few studies that have examined subtyping data and gene-level AMR data in raccoons [18, 19, 41, 42]; this study contributes new data to the literature concerning common serovars, sequence types, microbial population structure, resistance genes, and predicted plasmids carried by a rural raccoon population. In wild birds, where genomic investigations are becoming increasingly common, Enterobacteriales containing resistance genes to high-priority antimicrobials (e.g., $bla_{CTX-M-65}$, $bla_{IMP-4}$, *mcr-1*), and international clones have been identified [8, 43–45]. The resistance identified in raccoons and other sources on swine farms in our study mirrors that found in swine in other parts of the world (e.g., sulfonamides, aminoglycosides, tetracyclines) [46–48]. Our findings of widespread tetracycline resistance genes (*tetA*, *tetB*) that were not associated with particular sources, locations, or years are plausibly driven by the swine farm environment, since tetracyclines, among other antimicrobials, are among the more commonly

used antimicrobials in the Canadian swine industry [49, 50]. A previous study examining AMR in wild small mammals in a variety of environments in the same study region also found that resistance to tetracyclines (conferred by *tetA*) was significantly more likely to be identified in generic *E. coli* from animals trapped on swine farms compared to residential areas [51]. Overall, our findings are consistent with recent mounting evidence that the use of antimicrobials in agriculture is a major driver of AMR in intensive farm environments [52–54]. In the future, exploration of enteric bacteria and AMR carried by raccoons in different types of environments, including cities, may contribute valuable information concerning the impact of agricultural and urban environments on the microbiome of these animals.

## Limitations

The low overall prevalence of resistance among *Salmonella* isolates on swine farms presented an obvious challenge for the study of the movement of AMR determinants, and, in many cases, precluded statistical assessments of these data. Our low effective sample sizes did not permit multivariable modeling, or the ability to account for potential confounding by serovar (due to the sheer number of serovars identified, models could not converge). Our univariable analyses do not adequately capture the complexities of AMR in the ecosystem, but they represent an important first step in this process. Similarly, our analysis of Inc types should be interpreted with caution since we did not reconstruct plasmids or characterize mobility, thus, future work would be strengthened with confirmation of these aspects of plasmid biology [55]. Identification of the precise location of resistance genes either on plasmids or within chromosomes in future work will also help to provide further insights about AMR transmission and movement. As previously alluded to, the true relationships between predicted plasmids and the risk factors examined here are potentially obscured by our inclusion of only *E. coli* demonstrating phenotypic resistance. Moreover, inclusion of only one *E. coli* isolate per fecal sample limits our understanding of the transmission of AMR determinants within the greater gut microbiome [38], and of the true microbial population structure. The lack of similarities between *Salmonella* and *E. coli* obtained from the same animal therefore do not constitute definitive evidence that AMR gene transmission is not commonly occurring in the gastrointestinal tract of raccoons. Finally, the measures of association reported in our analyses do not account for the initial probability of isolating the organism in each sample. To illustrate this point, consider that the prevalence of resistant *E. coli* was highest in raccoons, and lowest in swine manure samples (35% vs 16%); the odds of certain predicted plasmids were higher in raccoons compared to swine manure isolates, but these odds were always relative to samples that already contained a resistant *E. coli* isolate.

## Conclusions

A diversity of *Salmonella* serovars were isolated from the raccoon population in this study, some of which have been implicated in human clinical cases in the study region (e.g., *S.* Thompson, *S.* Newport). Findings from our preliminary epidemiological investigation are suggestive of local transmission of certain strains of *Salmonella*, *E. coli*, and related AMR determinants between raccoons and environmental sources (i.e., soil, swine manure pits) locally on farms, and between farms in the region. Overall, our findings suggest that transmission of certain *Salmonella* serovars and related genes and plasmids is commonly occurring between soil and raccoons, but is rarely occurring between raccoons and swine manure pits. The highly variable distributions of resistance genes and predicted plasmids among different sources and locations revealed different epidemiological patterns for various AMR determinants in *Salmonella* and *E. coli*, highlighting the complexities underlying AMR transmission and

maintenance within the ecosystem. The integration of whole-genome sequence data within an epidemiological approach can help to guide and provide focus for future genomic investigations focused on transmission dynamics and phylogenetics. More comprehensive sampling of farm environments, and additional environmental sources, as well as a thorough examination of both susceptible and resistant *E. coli* isolates, is warranted.

## Supporting information

**S1 File. Gene accession numbers for *Salmonella* and *Escherichia coli* isolates.**
(XLSX)

**S1 Table. Multi-locus sequence types identified using whole-genome sequencing data from antimicrobial resistant *Escherichia coli* isolates obtained from raccoons, swine manure pits, and soil samples on swine farms in southern Ontario, Canada 2011–2013.**
(DOCX)

**S2 Table. Serovars identified using whole-genome sequencing data from phenotypically resistant *Escherichia coli* isolates obtained from raccoons, swine manure pits, and soil samples on swine farms in southern Ontario, Canada 2011–2013.**
(DOCX)

**S3 Table. Contrasts from univariable logistic regression models[a,b] (Table 6) assessing the statistically significant associations between farm location and the occurrence of select antimicrobial resistance genes and plasmid incompatibility (Inc) types in *Salmonella enterica* isolates from raccoons, swine manure pits, and soil samples on swine farms in southern Ontario, Canada 2011–2013.**
(DOCX)

**S4 Table. Contrasts from univariable logistic regression models[a,b] (Table 7) assessing the statistically significant associations between farm location, source type, and the occurrence of select antimicrobial resistance genes and plasmid incompatibility (Inc) types in *Escherichia coli* isolates from raccoons, swine manure pits, and soil samples on swine farms in southern Ontario, Canada 2011–2013.**
(DOCX)

## Acknowledgments

Erin Harkness, Samantha Kagan, and Mary Thompson assisted with collection of field data. Bryan Bloomfield collected swine manure pit samples and engaged the participation of landowners. Field samples were submitted by Tami Harvey and Barbara Jefferson. Culture of samples, and submission of samples for sequencing was performed and coordinated by the McEwen Lab at the Canadian Research Institute for Food Safety (Sarah Martz, laboratory technicians, and co-op students), University of Guelph, and the National Microbiology Laboratory at Guelph, Public Health Agency of Canada (OIE Reference Lab for Salmonellosis, Andrea Desruisseau, Chad Gill). This manuscript is dedicated to the memory of Sarah Martz (1988–2020).

## Author Contributions

**Conceptualization:** Nadine A. Vogt, Benjamin M. Hetman, David L. Pearl, Claire M. Jardine.

**Data curation:** Nadine A. Vogt, Adam A. Vogt, Kristin J. Bondo.

**Formal analysis:** Nadine A. Vogt, Benjamin M. Hetman, David L. Pearl, Adam A. Vogt.

**Funding acquisition:** David L. Pearl, Richard J. Reid-Smith, E. Jane Parmley, Michael R. Mulvey, Claire M. Jardine.

**Investigation:** Nadine A. Vogt, Nicol Janecko, Amrita Bharat, Kristin J. Bondo, Samantha E. Allen.

**Methodology:** Nadine A. Vogt, Benjamin M. Hetman.

**Project administration:** Nadine A. Vogt, Richard J. Reid-Smith, E. Jane Parmley, Michael R. Mulvey, Claire M. Jardine.

**Resources:** Nadine A. Vogt, Benjamin M. Hetman, Adam A. Vogt, Claire M. Jardine.

**Software:** Nadine A. Vogt, Benjamin M. Hetman, Adam A. Vogt.

**Supervision:** David L. Pearl, Richard J. Reid-Smith, E. Jane Parmley, Michael R. Mulvey, Nicole Ricker, Claire M. Jardine.

**Validation:** Nadine A. Vogt, Benjamin M. Hetman, David L. Pearl, Adam A. Vogt.

**Visualization:** Nadine A. Vogt, Benjamin M. Hetman.

**Writing – original draft:** Nadine A. Vogt.

**Writing – review & editing:** Nadine A. Vogt, Benjamin M. Hetman, David L. Pearl, Adam A. Vogt, Richard J. Reid-Smith, E. Jane Parmley, Nicol Janecko, Amrita Bharat, Michael R. Mulvey, Nicole Ricker, Kristin J. Bondo, Samantha E. Allen, Claire M. Jardine.

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
