## [Decision Letter · Decision Letter 0]

15 Oct 2021

PONE-D-21-30211Using whole-genome sequence data to examine the epidemiology of Salmonella, Escherichia coli and associated antimicrobial resistance in raccoons (Procyon lotor), swine manure pits, and soil samples on swine farms in southern Ontario, CanadaPLOS ONE

Dear Dr. Vogt,

Thank you for submitting your manuscript to PLOS ONE. After careful consideration, we feel that it has merit but does not fully meet PLOS ONE’s publication criteria as it currently stands. Therefore, we invite you to submit a revised version of the manuscript that addresses the points raised during the review process.

Please address all of the comments of the three reviewers, and particularly, the comments of reviewer #2.  This reviewer suggested comparing the data to a global Salmonella database to observe for broader global linkages, comparing phenotype to genotype calls, comparing the WGS data to the AMRFinderPlus database and uploading as metadata at AMRFinderPlus SRA, providing a list of all NCBI accession numbers for all genomes discussed, modifying Table 1 as suggested, providing detailed information for each unique isolate, providing phylogenetic tree testing, constructing a transmission network using StrainHub, and improving labeling of Figures.

We look forward to receiving your revised manuscript.

Kind regards,

Pina Fratamico, Ph.D.

Academic Editor

PLOS ONE

Journal Requirements:

"NAV received stipend funding through the Ontario Veterinary College, the University of Guelph, and a National Sciences and Engineering Research Council Postgraduate Scholarship-Doctoral."

"Funding was provided by the National Science and Engineering Research Council (NSERC) and the Ontario Ministry of Agriculture Food and Rural Affairs (OMAFRA). NAV received stipend funding through the Ontario Veterinary College, the University of Guelph, and a National Sciences and Engineering Research Council Postgraduate Scholarship-Doctoral. In-kind support was provided by Public Health Agency of Canada. The funders had no role in study design, data collection and analysis, decision to publish, or preparation of the manuscript. "

Reviewers' comments:

Reviewer's Responses to Questions

**Comments to the Author**

1. Is the manuscript technically sound, and do the data support the conclusions?

Reviewer #1: Yes

Reviewer #2: No

Reviewer #3: Yes

2. Has the statistical analysis been performed appropriately and rigorously? 

Reviewer #1: Yes

Reviewer #2: No

Reviewer #3: Yes

3. Have the authors made all data underlying the findings in their manuscript fully available?

Reviewer #1: Yes

Reviewer #2: No

Reviewer #3: No

4. Is the manuscript presented in an intelligible fashion and written in standard English?

Reviewer #1: Yes

Reviewer #2: Yes

Reviewer #3: Yes

5. Review Comments to the Author

Reviewer #1: The manuscript entitled “Using whole-genome sequence data to examine the epidemiology of Salmonella, Escherichia coli and associated antimicrobial resistance in raccoons (Procyon lotor), swine manure pits, and soil samples on swine farms in southern Ontario, Canada” presents a study in which whole genomes of Salmonella and antimicrobial resistant E. coli previously isolated from swine manure pits, raccoons, and soil in Ontario were analyzed and compared to assess transmission among these sources. Some overlapping bacterial sequence types, AMR genes, and plasmid types were identified in multiple sources and associated with particular sources or years. The authors did an excellent job presenting results and describing study limitations. This manuscript adds value to the field by both describing whole genome sequencing data of antimicrobial resistant bacteria found in wildlife and the environment and by assessing epidemiological links based on those data. I have a few relatively minor comments discussed below.

Lines 157-159: Please discuss selection of k values here and why different values were selected for Salmonella and E. coli. Also specifically state that these trees were based on cgMLST.

Lines 273-275: What about E. coli sequence types commonly found in humans (e.g. UPEC strains)?

GrapeTree figures. Please include in the figure legend what K means. (i.e. for E. coli, each circle includes isolates differing by no more than 50 alleles).

Line 503: The term “extensively drug resistant” has precise definitions (e.g. Magiorakos et al., 2012, https://www.clinicalmicrobiologyandinfection.com/article/S1198-743X(14)61632-3/fulltext). Please check the use of this term and provide a reference for the definition used here.

Lines 510-521: This is not new information and was already published and discussed in Bondo et al., 2016. Please remove or discuss how WGS data influenced this assessment.

Lines 541-544: Many of the AMR genes found are very common in other environments and hosts. I suggest including an alternative explanation (e.g. widespread presence of these genes in other sources). Additionally, “Or the presence of widespread closely related isolates” is not a very plausible explanation given that you identified many (49) different E. coli sequence types.

Lines 606-608: Are the results suggestive of transmission among those three sources? What about other sources not sampled? Given the high diversity of STs and what it stated in Lines 611-615.

Reviewer #2: Line 101 “For the present study, we included only isolates originating from swine farm environments” Authors should consider wide-spread comparison of global samples by uploading and comparing their data to a global salmonella database like NCBI Pathogen detection where they might see broader global linkages.

Line 129-130, 184, 319 Authors should directly compare phenotype to genotype calls and report any discrepancies. All WGS data should be compared to AMRFinderPlus assessment of AMR gene presence which provides AMR for more then 15 phenotypically determined antibiograms. Additionally, all Antibiograms should be uploaded as metadata at AMRFinderPlus SRA and biosample so that this phenotype to genotype data can be cataloged to improve future AMR predictions. Incongruences were partly discussed line 380 but should be expanded to explicitly discuss all failed predictions as these represent new AMR genes or alleles that are not resistant.

Table 1. Data availability and public release of data. The authors are encouraged to provide a list of all NCBI accession numbers for all genomes discussed in the manuscript. The investigators should provide two additional columns to Table 1. One for strain identification and a second for NCBI accession number of the WGS data per isolate. The authors could also add to the list stress and pathogenicity genes made available at NCBI AMRFinderPlus once the data has been publicly released. They should release the data now to add these results to their manuscript. The authors should consider listing each unique isolate included in their study and include all detailed information for each isolate. Just providing aggregated results is insufficient to reproduce their results.

Line 231 As with the Salmonella, all E. coli Strain identification and NCBI accession numbers should also be added to Table 1 for full transparency for all isolates included in the study.

Line 264, 273. If the authors release all of their data publicly at the NCBI Pathogen Detection web site then they can see if their new WGS data clusters with any known clinical isolate rather than speculating on the possibility they can see if any direct clusters exist in the public release genomes for Salmonella and E. coli. The authors are advised to publicly release their data and make these comparisons and report the results.

Line 172 Using the snippy results the authors are recommended to provide phylogenetic tree testing whether any of these isolates cluster with any others. Independently they can see what clusters at NCBI Pathgen detection.

Line 446-447 To study the pathogen transmission dynamics the authors should consider constructing a transmission network using StrainHub, version 0.2.0. This is a phylogenetic approach to understanding transmission dynamics.

Figures. The investigators have not labeled many of the terminal nodes in their figures with the strain identification so none of these figures is interpretable to the detail of what is claimed in the manuscript. All terminal nodes should be labeled with a unique strain ID.

Table 1. The investigators have not provided NCBI Accession numbers for the genomes described in this study and so none of the Data is currently available publicly released so I am unable to test any of the claims made by these authors. All data must be publicly released for PlosOne to promote openness and transparency to advance science.

Reviewer #3: 1. I did not see the big project numbers, Sequence Read Archive (SRA) numbers or gene accession numbers for whole genome sequencing data. Where were the data deposited?

2. There were 3 Salmonella and 2 E. coli isolates not typerable by MLST. Please provide an explanation for this.

3. Please explain why 15 antibiotics were chosen for this study and why fosfomycin was not included in the panel

6. PLOS authors have the option to publish the peer review history of their article (what does this mean?). If published, this will include your full peer review and any attached files.

Reviewer #1: No

Reviewer #2: No

Reviewer #3: No

---

## [Author Response · Author response to Decision Letter 0]

2 Nov 2021

Response to Reviewers

We thank all reviewers and the editor for their helpful feedback to improve the manuscript. Our responses are below. Line numbers correspond to tracked changes version. 

Please address all of the comments of the three reviewers, and particularly, the comments of reviewer #2. This reviewer suggested comparing the data to a global Salmonella database to observe for broader global linkages, comparing phenotype to genotype calls, comparing the WGS data to the AMRFinderPlus database and uploading as metadata at AMRFinderPlus SRA, providing a list of all NCBI accession numbers for all genomes discussed, modifying Table 1 as suggested, providing detailed information for each unique isolate, providing phylogenetic tree testing, constructing a transmission network using StrainHub, and improving labeling of Figures.

We have modified Table 1 as suggested, and also added a supplementary table with a list of all NCBI accession numbers for all genomes included (File S1). We appreciate the suggestions for potential avenues for analyses using a fine-grained genomics approach. However, our goal in using WGS data was to integrate these data into a broader traditional epidemiological approach. Moreover, in the interest of keeping our manuscript streamlined and focused on the broader epidemiological assessment which seeks to characterize population-level patterns and distributions of genes/plasmids among different sources, we have opted to keep our existing analyses as they are. We feel that the addition of further analyses with a different aim would overcomplicate the manuscript and obscure the focus, which is predominantly on source-level (not strain-level) transmission, and on the overarching epidemiological patterns of AMR determinants. The use of WGS within traditional epidemiological frameworks is an emerging field, and there is limited available literature combining these approaches (mostly due to limited sample sizes for statistical analyses), thus, the value of this manuscript lies in the use of WGS-derived outputs within a traditional epidemiological analysis. Overall, we see many of these suggestions as valuable avenues for future work focusing on the strain-level aspects of this population of isolates, and with open data we encourage other researchers to build on this work using a finer-grained genomics-focused approach. 

In addition, we are currently working on submitting a related manuscript comparing the Salmonella isolates from this previous wildlife study with clinical human and livestock isolates obtained from the same geographic region and time period. We agree that performing a global assessment and comparison of the farm-level isolates in this manuscript to other external sources would be a valuable assessment, however, inclusion of this type of analysis in the present manuscript would tread on the work performed in our manuscript that is in preparation. In order to make clear our epi-based approach for the reader, we have modified the first line of the discussion to better situate this work in the literature, as primarily a source-level epidemiological assessment. Thus, phylogenetic tree testing to determine evolutionary history and construction of detailed transmission networks are beyond the scope of the current manuscript. 

Since the figures were only provided as a means of qualitatively displaying the population structure and overlap of different organisms between different sources, we maintain that it would impede interpretation of the figures if additional strain level information was added (since our sample sizes are considerable---n=159 and n=96 for Salmonella and E. coli, respectively). All comparisons made in text were performed by comparing cgMLST types in R, and were not performed using GrapeTree. We have added text to the methods to convey the purpose of creating minimum spanning trees (lines 161-175). 

Our sequence data were submitted to Genbank on July 26, 2021, and were scheduled to be released on September 30, 2021, but I just confirmed with them that, unfortunately, due to a backlog, the data were only recently released, on October 29, 2021. Please see BioProject number PRJNA745182. 

Our results can be replicated by linking the sequence data and strain numbers with the epidemiological data previously released by Bondo et al. We hesitate to add in accession numbers into the text for the section in between lines 300-312, since listing up to 29 accession numbers, for example, would severely hinder the communication of these findings in text. These analyses can easily be replicated in R, as we have specified which serovars and sources were examined, and there are a limited number of isolates with those epidemiological attributes.

Reviewer #1: The manuscript entitled “Using whole-genome sequence data to examine the epidemiology of Salmonella, Escherichia coli and associated antimicrobial resistance in raccoons (Procyon lotor), swine manure pits, and soil samples on swine farms in southern Ontario, Canada” presents a study in which whole genomes of Salmonella and antimicrobial resistant E. coli previously isolated from swine manure pits, raccoons, and soil in Ontario were analyzed and compared to assess transmission among these sources. Some overlapping bacterial sequence types, AMR genes, and plasmid types were identified in multiple sources and associated with particular sources or years. The authors did an excellent job presenting results and describing study limitations. This manuscript adds value to the field by both describing whole genome sequencing data of antimicrobial resistant bacteria found in wildlife and the environment and by assessing epidemiological links based on those data. I have a few relatively minor comments discussed below.

Lines 157-159: Please discuss selection of k values here and why different values were selected for Salmonella and E. coli. Also specifically state that these trees were based on cgMLST.

Thank you for the suggestion. We have specified our rationale and approach to selection of k values for the different figures, and we have also clarified in text that the trees were based on cgMLST (lines 161-175). 

Lines 273-275: What about E. coli sequence types commonly found in humans (e.g. UPEC strains)?

Thank you for the suggestion. A statement has been added in text (lines 293-295) and a footnote to Table S1. 

GrapeTree figures. Please include in the figure legend what K means. (i.e. for E. coli, each circle includes isolates differing by no more than 50 alleles).

All figure headings have been modified as suggested. 

Line 503: The term “extensively drug resistant” has precise definitions (e.g. Magiorakos et al., 2012, https://www.clinicalmicrobiologyandinfection.com/article/S1198-743X(14)61632-3/fulltext). Please check the use of this term and provide a reference for the definition used here.

Thank you for catching this error. We have corrected the text and used the term MDR instead (line 531). 

Lines 510-521: This is not new information and was already published and discussed in Bondo et al., 2016. Please remove or discuss how WGS data influenced this assessment.

Thank you for catching the redundancy. These lines have been removed. 

Lines 541-544: Many of the AMR genes found are very common in other environments and hosts. I suggest including an alternative explanation (e.g. widespread presence of these genes in other sources). Additionally, “Or the presence of widespread closely related isolates” is not a very plausible explanation given that you identified many (49) different E. coli sequence types.

We agree this wasn't the most plausible explanation. The text has been changed to reflect that the most likely alternative explanation was "common exposure to AMR pollution in environmental sources". (lines 573-574). 

Lines 606-608: Are the results suggestive of transmission among those three sources? What about other sources not sampled? Given the high diversity of STs and what it stated in Lines 611-615.

Thank you for catching the apparent contradiction in our conclusion. We have removed the text that was contradictory and substituted a summary statement conveying that the epidemiological pattern of each AMR determinant was variable (lines 639-651). We maintain that there does appear to be evidence suggestive of transmission for certain strains and serovars, but not all (line 637). 

Reviewer #2: Line 101 “For the present study, we included only isolates originating from swine farm environments” Authors should consider wide-spread comparison of global samples by uploading and comparing their data to a global salmonella database like NCBI Pathogen detection where they might see broader global linkages.

See above comments. We agree that a comparison of these isolates to a global database would be useful and interesting, however, the focus of this manuscript was to use an epi-based approach with WGS-derived data originating from swine farm environments specifically. Our related manuscript that is currently in preparation addresses Salmonella and related AMR transmission between these swine farm isolates with human clinical cases and livestock isolates from the same geographic region and time period. We are cautious about keeping the objectives of the two manuscripts distinct. 

Line 129-130, 184, 319 Authors should directly compare phenotype to genotype calls and report any discrepancies. All WGS data should be compared to AMRFinderPlus assessment of AMR gene presence which provides AMR for more then 15 phenotypically determined antibiograms. Additionally, all Antibiograms should be uploaded as metadata at AMRFinderPlus SRA and biosample so that this phenotype to genotype data can be cataloged to improve future AMR predictions. Incongruences were partly discussed line 380 but should be expanded to explicitly discuss all failed predictions as these represent new AMR genes or alleles that are not resistant.

Since we have so many isolates (n=255), our objective in performing a class-level (not gene-level) assessment of genotype to phenotype was to provide a course-grained, succinct assessment to validate our epidemiological analyses. In line with our objectives, isolates with too many failed predictions were excluded from statistical analyses to remove potentially unreliable information from our analyses. As such, we prioritized population-level analyses (in line with our epidemiological approach), rather than delving into missed predictions of individual isolates and discovery of new AMR genes. Resfinder and ABricate use a subset of the AMR genes in the AMRFinderPlus database, and cover all of the antimicrobials (and more!) that were previously tested phenotypically for in previous work by Bondo et al. Thus, we consider the use of Resfinder to be more than adequate for the scope of our paper. 

Although we would not be opposed to depositing our sequence data to an additional repository to promote scientific advancements, this process represents a considerable time and financial burden to the primary author (i.e., unpaid work requiring access to high-speed internet, currently unavailable in the primary author's residence, and no access to university facilities). This request also goes above and beyond what is considered "Open Science" (and beyond what is required by Plos One for publishing). The phenotypic data are available from Bondo et al.'s previous release of data, and our sequence data are available from Genbank. We have provided all data needed for another researcher to perform gene-level comparisons of phenotype to genotype, if desired. Finally, the inclusion of redundant analyses (i.e., both class-level and gene-level comparisons) has the potential to overcomplicate the paper, since we already have three figures, seven tables, and four supplementary tables. 

Table 1. Data availability and public release of data. The authors are encouraged to provide a list of all NCBI accession numbers for all genomes discussed in the manuscript. 

A supplementary table has been added with these accession numbers (S1 File). 

The investigators should provide two additional columns to Table 1. One for strain identification and a second for NCBI accession number of the WGS data per isolate. 

Thank you for the suggestions, these have been added to the table 1. 

The authors could also add to the list stress and pathogenicity genes made available at NCBI AMRFinderPlus once the data has been publicly released. They should release the data now to add these results to their manuscript. 

Thank you for the suggestion, however, we feel these approaches are outside the scope of our epidemiologically-focused investigation looking at source-level transmission (see newly added lines 470-473). As of October 29, 2021 our data is available in Genbank if other researchers wish to pursue this research angle. 

The authors should consider listing each unique isolate included in their study and include all detailed information for each isolate. Just providing aggregated results is insufficient to reproduce their results.

We have a total of 255 isolates included in this study, thus we feel that including a table with 255 lines is not very interpretable for the reader, given that we already have three figures, seven tables, and four supplementary tables. Using a population-level (epidemiological) approach necessitates presentation of aggregate results; this is particularly true of all of the statistical analyses. Traditional epidemiological analyses rarely present raw data, since the goal of the analysis is to provide an assessment of a large number of samples/isolates that cannot be assessed via qualitative examination alone. Providing our sequence data, associated epidemiological data, and methodological approaches is sufficient to reproduce our results. 

Line 231 As with the Salmonella, all E. coli Strain identification and NCBI accession numbers should also be added to Table 1 for full transparency for all isolates included in the study.

The purpose of Table 1 was to present the antimicrobial resistant Salmonella isolates within the population, not the E. coli, since all 96 of those isolates were selected based on the presence of their phenotypic resistance. 

Line 264, 273. If the authors release all of their data publicly at the NCBI Pathogen Detection web site then they can see if their new WGS data clusters with any known clinical isolate rather than speculating on the possibility they can see if any direct clusters exist in the public release genomes for Salmonella and E. coli. The authors are advised to publicly release their data and make these comparisons and report the results. 

As previously mentioned, the objective of our epidemiologically-focused paper was to assess for transmission on swine farms, and not compare to a larger database, or other sources---this is the focus of our related manuscript that is currently in preparation. We insist on keeping the focus of these two papers distinct. 

Line 172 Using the snippy results the authors are recommended to provide phylogenetic tree testing whether any of these isolates cluster with any others. Independently they can see what clusters at NCBI Pathgen detection.

Unfortunately, we were limited by computational power, thus the use of cgMLST was preferred, and we consider it sufficient for our epidemiological approach (additionally, this typing method is widely used by PulseNet for foodborne illness outbreak investigations, see our reference in the manuscript, Tolar et al., 2019). We were only able to perform a SNP-based approach to provide higher resolution for a small subset of specific isolates of interest. As we highlighted in the comments above, our approach was predominantly focused on source-level transmission using an epidemiological approach, rather than a specific genomics-based approach focused on evolutionary history and overlap with sources outside of the swine farm environment. As the GrapeTree documentation highlights, "It is difficult to infer clusters from classical phylograms when showing large numbers of isolates", as we have here. Furthermore, the primary author is currently limited by computational power to use SNIPPY to assess 159 and 96 isolates, respectively. A preliminary SNP-based assessment using Roary and FastTree was attempted in 2020, but the risk to the computer's hardware was too high to reattempt (so much power was needed that the computer died even though it was plugged into the wall outlet), and to consider correcting for potential confounding due to recombination (essential for any epidemiological analyses using WGS data) using alternative software to FastTree would require even further computational power. 

Line 446-447 To study the pathogen transmission dynamics the authors should consider constructing a transmission network using StrainHub, version 0.2.0. This is a phylogenetic approach to understanding transmission dynamics.

We maintain that keeping our approach simple, and as a preliminary "scan" of the data using epidemiologically-informed statistical modeling, will help to lay the groundwork for future research using more complex genomic approaches, and allow researchers to focus-in on subsets of the isolates meriting further investigation. As such, these types of phylogenetic analyses were outside the scope of this paper, but will certainly be important in future papers with a different aim. 

Figures. The investigators have not labeled many of the terminal nodes in their figures with the strain identification so none of these figures is interpretable to the detail of what is claimed in the manuscript. All terminal nodes should be labeled with a unique strain ID.

See above comment: since the figures were only provided as a means of qualitatively displaying the population structure and overlap of different organisms between different sources, we maintain that there is no need to clutter the figures with additional strain level information (since our sample sizes are considerable---n=159 and n=96 for Salmonella and E. coli, respectively). All comparisons made in text were performed by comparing cgMLST types of similar isolates and were not performed using GrapeTree. We have added text to the methods to convey the purposes of creating minimum spanning trees (lines 161-175). 

Table 1. The investigators have not provided NCBI Accession numbers for the genomes described in this study and so none of the Data is currently available publicly released so I am unable to test any of the claims made by these authors. All data must be publicly released for PlosOne to promote openness and transparency to advance science.

The data were submitted to Genbank on July 26, 2021, and were to be released on September 30, 2021, but I just confirmed with them that due to a backlog, the data were not released until October 29, 2021. They are available now. See project number PRJNA745182 in Genbank. We have added a supplementary file (File S1) with information about all isolates included in the study, with the following attributes: the bioproject number, accession ids, biosample numbers, sample ids, sequencing ids, sampling sources, collection date, and bacterial species. 

Reviewer #3: 1. I did not see the big project numbers, Sequence Read Archive (SRA) numbers or gene accession numbers for whole genome sequencing data. Where were the data deposited?

Apologies that the data were not available at the time of submission. The WGS data were supposed to be publicly released on September 30, 2021 under Bioproject # PRJNA745182, but I've contacted Genbank and due to a backlog the data weren't released until October 29, 2021. We have included a list of accession numbers for the isolates included in our work in S1 File, along with other attributes (see comment above). 

2. There were 3 Salmonella and 2 E. coli isolates not typeable by MLST. Please provide an explanation for this.

The isolates that were not typeable were assigned a "—" by the MLST program. None of these isolates were assigned a MLST type because at least one allele was missing, or there was only a partial match. We have added text to clarify this (lines 259-262; 290-291). 

3. Please explain why 15 antibiotics were chosen for this study and why fosfomycin was not included in the panel

The phenotypic assessment performed was part of previous work using the standard CIPARS or NARMS panel of antimicrobials, which, unfortunately, does not include fosfomycin. Unfortunately, our funding did not permit us to revisit phenotypic microbial assessments from previous work based on our WGS genotypic AMR findings. We have added the clarification in text that this work was previously performed (lines 117, 118, and 127).

---

## [Editor Report · Decision Letter 1]

5 Nov 2021

Using whole-genome sequence data to examine the epidemiology of Salmonella, Escherichia coli and associated antimicrobial resistance in raccoons (Procyon lotor), swine manure pits, and soil samples on swine farms in southern Ontario, Canada

PONE-D-21-30211R1

Dear Dr. Vogt,

We’re pleased to inform you that your manuscript has been judged scientifically suitable for publication and will be formally accepted for publication once it meets all outstanding technical requirements.

Kind regards,

Pina Fratamico, Ph.D.

Academic Editor

PLOS ONE
---

## [Editor Report · Acceptance letter]

9 Nov 2021

PONE-D-21-30211R1 

Using whole-genome sequence data to examine the epidemiology of Salmonella, Escherichia coli and associated antimicrobial resistance in raccoons (Procyon lotor), swine manure pits, and soil samples on swine farms in southern Ontario, Canada 

Dear Dr. Vogt:

I'm pleased to inform you that your manuscript has been deemed suitable for publication in PLOS ONE. Congratulations! Your manuscript is now with our production department. 

Kind regards, 

on behalf of

Dr Pina Fratamico 

Academic Editor

PLOS ONE